# LEARNING DYNAMICAL SYSTEMS WITH HELMHOLTZ-HODGE DECOMPOSITION AND GAUSSIAN PROCESSES

## ABSTRACT

Machine learning models provide alternatives for efficiently recognizing complex patterns from data, but two main concerns in applying them to modeling physical systems stem from their physics-agnostic design and lack of interpretability. This paper mitigates these concerns by encoding the Helmholtz-Hodge decomposition into a Gaussian process model, leading to a versatile framework that simultaneously learns the curl-free and divergence-free components of a dynamical system. Learning a predictive model in this form facilitates the exploitation of symmetry priors. In addition to improving predictive power, these priors link the identified features to comprehensible scientific properties of the system, thus complex responses can be modeled while retaining interpretability. We show that compared to baseline models, our model achieves better predictive performance on several benchmark dynamical systems while allowing accurate estimation of the energy evolution of the systems from noisy and sparse data.

## 1 INTRODUCTION

A dynamical system describes how the state of a system evolves over time (Strogatz, 2018). Data-driven modeling of a dynamical system has become a fundamental task in many modern science and engineering applications, including physical emulation (Garnett et al., 2015) and robotics control (Deisenroth et al., 2015). Mathematically, a dynamical system is often a set of first-order ordinary differential equations (ODEs) or, equivalently, a smooth vector field on a manifold (Hirsch et al., 2012). Given the functional form of ODEs, classical data-driven methods typically involve optimizing their parameters (Ramsay et al., 2007). However, for many complex systems it is practically difficult to determine the form of the equations governing the underlying dynamics.

Recent advances in machine learning (ML) focus on the use of neural networks (Chen et al., 2018) and nonparametric Bayesian models (Solak et al., 2002; Heinonen et al., 2018; Hegde et al., 2022) for the black-box approximation of vector fields. Although these models have rich expressive power, there are two fundamental challenges in applying them to the modeling of dynamical systems. The first concern is related to their lack of interpretability. Being able to explain what a model has learned is extremely useful because it allows us to gain insight into the behavior of a dynamical system. However, it is often unclear how to transfer the features identified by the ML models to comprehensible scientific properties. Second, predictions from ML models in their native forms are prone to violating physical laws. To address these issues, a popular approach is to develop models that incorporate strong physical priors as inductive biases. Such prior knowledge commonly stems from basic physical principles related to certain differential invariants of vector fields. For example, in scenarios of learning Hamiltonian systems (Greydanus et al., 2019; Toth et al., 2019; Rath et al., 2021) and incompressible fluid dynamics (Wandel et al., 2021; Kiessling et al., 2021), ML models are constructed to learn divergence-free (div-free) vector fields, as a consequence of conservation laws of energy or mass. These powerful physical principles effectively improve the extrapolation performance of the ML models, but they limit the application scope of the models. For example, a div-free vector field fails to describe a dynamical system with dissipation, but real-world dynamical systems always suffer from non-negligible dissipation.

To develop a predictive model covering more dynamical systems, we explore supplementing the div-free vector field with a curl-free vector field. This is inspired by the Helmholtz-Hodge decomposition (HHD) (Arfken & Weber, 1999; Majda & Bertozzi, 2001; Bhatia et al., 2012), which states

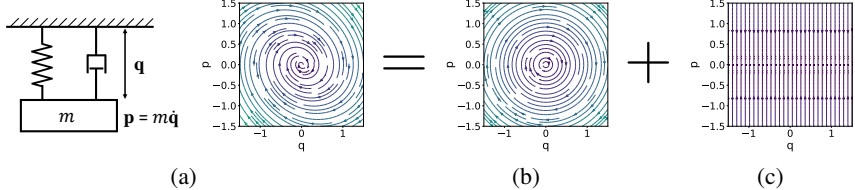

(a)                    (b)                    (c)

Figure 1: The vector field of a damped mass-spring system (a) can be decomposed into a div-free component (b) and a curl-free component (c), the former representing its Hamiltonian dynamics and the latter describing the friction-induced dissipation. Color indicates the magnitude of the vectors.

that any sufficiently smooth vector field can be expressed as the sum of a curl-free vector field and a div-free vector field. HHD is widely used in the study of Navier-Stokes equations (Maria Denaro, 2003; Garba & Haldenwang, 2013; Caltagirone, 2021), but in this work we explore its connections with more general dynamical systems. For example, as shown in Fig. 1, HHD can be used to characterize the dynamics of a dissipative Hamiltonian system (see Appendix A.1 for more details). Some existing works (Greydanus & Sosanya, 2022; Desai et al., 2021; Bhattoo et al., 2023) learn dissipative dynamics by compensating Lagrangian or Hamiltonian NNs with dissipative terms. However, these model structures involving physical governing equations may limit their scope of application. For example, they cannot be applied to the Chua circuit system, a chaotic system that is difficult to describe using Hamiltonian or Lagrangian equations. In contrast, we build a model from the perspective of satisfying certain differential invariants (either free of divergence or of curl), which thus allows for more application scenarios.

In this work, we construct Gaussian process (GP) priors for div-free and curl-free dynamics separately, resulting in an additive GP model in the form of HHD for learning vector fields of dynamical systems. Our resulting HHD-GP allows one to leverage more prior knowledge than modeling the system as a whole. In particular, we investigate its potential in exploiting priors of symmetries, motivated by the observation that the div-free and curl-free components of a dynamical system usually exhibit more symmetry properties than the system itself. For example, the damped mass-spring system in Fig. 1a exhibits odd symmetry, but its div-free (Fig. 1b) and curl-free (Fig. 1c) components additionally present rotation and translation symmetry, respectively. Therefore, we further build symmetry-preserving div-free GPs and curl-free GPs by exploiting closure of GPs under linear transformation. The symmetry prior not only improves the predictive performance of HHD-GP, but also makes it identifiable, thus identified div-free and curl-free features can be physically-meaningful. In particular, by exploiting the connection between HHD and the generalized Hamiltonian formalism, the learned div-free features are closely related to the energy of the dynamical systems.

The main contributions of this work are summarized as follows:

- We introduce a GP prior (called HHD-GP) for learning the Helmholtz-Hodge decomposition of an $n$-dimensional dynamical system.
- We construct a symmetry-preserving extension of the HHD-GP that can learn physically-meaningful representations of dynamical systems.
- Experiments on several dissipative systems show that our model can both accurately predict their dynamics and energy evolution from sparse, noisy observations.

## 2 BACKGROUND

### 2.1 HHD AND PROBLEM SETUP

We consider an autonomous system governed by the following ODEs:

$$\dot{\mathbf{x}}(t) := \frac{d\mathbf{x}(t)}{dt} = \mathbf{f}(\mathbf{x}(t)) = \mathbf{f}_{curl}(\mathbf{x}(t)) + \mathbf{f}_{div}(\mathbf{x}(t)), \tag{1}$$

which defines a vector field by assigning a vector $\mathbf{f}(\mathbf{x}) \in \mathbb{R}^n$ to every state $\mathbf{x} \in \mathbb{R}^n$. We assume that the vector field $\mathbf{f} \in L^2(\mathbb{R}^n, \mathbb{R}^n)$ is smooth, and any such vector field can be decomposed into the sum of a curl-free vector field $\mathbf{f}_{curl} : \mathbb{R}^n \to \mathbb{R}^n$ ($\nabla \wedge {}^1\mathbf{f}_{curl} = \mathbf{0}, \forall \mathbf{x} \in \mathbb{R}^n$) and a divergence-free vector field $\mathbf{f}_{div} : \mathbb{R}^n \to \mathbb{R}^n$ ($\nabla \cdot \mathbf{f}_{div} = 0, \forall \mathbf{x} \in \mathbb{R}^n$), according to the *Helmholtz–Hodge*

---

${}^1\wedge$ is the geometric outer product.

*decomposition* (HHD) (Majda & Bertozzi, 2001; Bhatia et al., 2012). In this work, we are interested in learning $\mathbf{f}$, $\mathbf{f}_{curl}$ and $\mathbf{f}_{div}$ simultaneously from a collection of noisy observations denoted by $\mathcal{D} = \{(\mathbf{x}_i, \mathbf{y}_i)\}_{i=1}^m$, with a noisy observation $\mathbf{y}_i$ at a state $\mathbf{x}_i$ given by

$$\mathbf{y}_i = \mathbf{f}(\mathbf{x}_i) + \epsilon, \epsilon \overset{\text{i.i.d}}{\sim} \mathcal{N}(\mathbf{0}, \Omega), \tag{2}$$

where an additive noise $\epsilon \in \mathbb{R}^n$ follows a zero-mean Gaussian distribution defined by a covariance matrix $\Omega = \text{diag}\left(\sigma_1^2, \ldots, \sigma_n^2\right) \in \mathbb{R}^{n \times n}$ modeling noise variance in each output dimension.

## 2.2 VECTOR-VALUED GP MODEL

We are interested in using Gaussian processes (GPs) to infer unknown vector fields. A GP is a stochastic process commonly used as a distribution for functions, assuming that any finite number of function values has a joint Gaussian distribution (Rasmussen & Williams, 2005). To learn an unknown vector field $\mathbf{f} : \mathbb{R}^n \to \mathbb{R}^n$, we assume a vector-valued GP prior as $\mathbf{f}(\mathbf{x}) \sim \mathcal{GP}(\mathbf{0}, \kappa(\mathbf{x}, \mathbf{x}'))$, where the mean of the function values is set to zero, and the covariance is captured by a matrix-valued kernel $\kappa : \mathbb{R}^n \times \mathbb{R}^n \to \mathbb{R}^{n \times n}$, whose $(i, j)$-th entry expresses the correlation between the $i$-th dimension of $\mathbf{f}(\mathbf{x})$ and the $j$-th dimension of $\mathbf{f}(\mathbf{x}')$. In the GP framework, the kernel controls the properties of possible functions under a GP, leading to various efforts of problem-specific design of kernels (Wilson & Adams, 2013; Durrande et al., 2012; Duvenaud, 2014). And to be a valid covariance function, the kernel should be symmetric and positive semidefinite (Álvarez et al., 2012).

A GP provides a Bayesian non-parametric approach for solving regression tasks. According to the GP prior, function values at inputs $\mathbf{X} = [\mathbf{x}_1, \ldots, \mathbf{x}_m]^\mathsf{T}$ are jointly distributed as $\mathcal{N}(\mathbf{f}(\mathbf{X}); \mathbf{0}, \mathbf{K})$, where $\mathbf{K} = [\kappa(\mathbf{x}_i, \mathbf{x}_j) \in \mathbb{R}^{n \times n}]_{i,j=1}^m$ is a block-partitioned covariance matrix. Then the marginal likelihood of the noisy observations $\mathbf{Y} = [\mathbf{y}_1, \ldots, \mathbf{y}_m]^\mathsf{T}$ given by Eq. 2 can be calculated by

$$p(\mathbf{Y} \mid \mathbf{X}) = \int p(\mathbf{Y} \mid \mathbf{X}, \mathbf{f}(\mathbf{X})) p(\mathbf{f}(\mathbf{X}) \mid \mathbf{X}) d\mathbf{f} = \mathcal{N}(\mathbf{Y} \mid \mathbf{0}, \mathbf{K} + \Sigma), \tag{3}$$

where $\Sigma = \Omega \otimes I_m$ is a diagonal matrix whose elements are the variance of the observation noise. Training a GP model often refers to maximizing the log of Eq. 3 to optimize the kernel parameters and the noise variance. Then by conditioning on these observations using Bayes' rule, the predictive posterior for a new state $\mathbf{x}_*$ is still Gaussian with its mean $\mu$ and variance $v$ given by

$$\mu(\mathbf{x}_*) = \mathbf{k}^\mathsf{T}(\mathbf{K} + \Sigma)^{-1}\mathbf{Y}, \; v(\mathbf{x}_*) = \kappa(\mathbf{x}_*, \mathbf{x}_*) - \mathbf{k}^\mathsf{T}(\mathbf{K} + \Sigma)^{-1}\mathbf{k}, \tag{4}$$

where $\mathbf{k} = [\kappa(\mathbf{x}_1, \mathbf{x}_*), \ldots, \kappa(\mathbf{x}_m, \mathbf{x}_*)]^\mathsf{T} \in \mathbb{R}^{mn \times n}$. The derived mean function is used for regression results, and the associated variance quantifies prediction uncertainty. Due to their nonparametric nature, GPs can self-adapt to the complexity of the target function based on the data provided, without being restricted to specific parametric forms.

## 3 HHD-GP MODEL

We consider the problem of learning a continuous-time dynamical model in the form of HHD (Eq. 1) with GPs. HHD points out the prevalence of an additive structure in dynamical systems, so the key idea here is to exploit two GPs to model $\mathbf{f}_{curl}$ and $\mathbf{f}_{div}$ respectively,

$$\mathbf{f}_{curl} \sim \mathcal{GP}(\mathbf{0}, \kappa_{curl}(\mathbf{x}, \mathbf{x}')), \; \mathbf{f}_{div} \sim \mathcal{GP}(\mathbf{0}, \kappa_{div}(\mathbf{x}, \mathbf{x}')). \tag{5}$$

Then the sum of these two GPs results in a new GP modeling the dynamical system $\mathbf{f}$, with a new kernel function defined as the sum of the curl-free and divergence-free ones. And the additive kernel $\kappa_{hdd} = \kappa_{curl} + \kappa_{div}$ inherits the symmetric and positive-semidefinite properties of $\kappa_{curl}$ and $\kappa_{div}$, so the GP predictor (Eq. 4) is valid for the additive GP model. And the additivity of the kernels implies the additivity of GP means, so the mean function $\mu(\mathbf{x}_*)$ in Eq. 4 can be split into a curl-free part $\mu_{curl}(\mathbf{x}_*)$ and a divergence-free part $\mu_{div}(\mathbf{x}_*)$, and we have

$$\mu(\mathbf{x}_*) = \mathbf{k}_{curl}^\mathsf{T}\mathbf{K}_*^{-1}\mathbf{Y} + \mathbf{k}_{div}^\mathsf{T}\mathbf{K}_*^{-1}\mathbf{Y} = \mu_{curl}(\mathbf{x}_*) + \mu_{div}(\mathbf{x}_*), \tag{6}$$

where $\mathbf{K}_* = \mathbf{K}_{curl} + \mathbf{K}_{div} + \Sigma$. It can be seen that the effects of $\mathbf{f}_{curl}$ and $\mathbf{f}_{div}$ can be treated as observation noises for each other, so their prediction variances at $\mathbf{x}_*$ are obtained by

$$v_{curl}(\mathbf{x}_*) = \kappa_{curl}(\mathbf{x}_*, \mathbf{x}_*) - \mathbf{k}_{curl}^\mathsf{T}\mathbf{K}_*^{-1}\mathbf{k}_{curl}, \; v_{div}(\mathbf{x}_*) = \kappa_{div}(\mathbf{x}_*, \mathbf{x}_*) - \mathbf{k}_{div}^\mathsf{T}\mathbf{K}_*^{-1}\mathbf{k}_{div}. \tag{7}$$

Consequently, observations of $\mathbf{f}(\mathbf{x})$ given by Eq. 2 can be used to make predictions for its hidden components $\mathbf{f}_{curl}$ and $\mathbf{f}_{div}$. So now our goal is to construct GPs with realizations in the space of curl-free and div-free vector fields.

In the following parts, we construct the kernels $\kappa_{curl}$ and $\kappa_{div}$ from the representations of $\mathbf{f}_{curl}$ and $\mathbf{f}_{div}$ respectively, exploiting closure of GPs under linear transformation. Let $R = \mathbb{R}\left[\partial_{x_1}, \ldots, \partial_{x_n}\right]$ be the polynomial ring in the partial derivatives, and $\mathcal{L} \in R^{a \times b}$ be a matrix of differential operators acting on functions $g : \mathbb{R}^n \to \mathbb{R}^b$ distributed as $\mathcal{GP}\left(\mu\left(\mathbf{x}\right), \kappa\left(\mathbf{x}, \mathbf{x}'\right)\right)$. Then, the transformation of $g$ under $\mathcal{L}$ is again distributed as a GP with

$$\mathcal{L}g \sim \mathcal{GP}\left(\mathcal{L}\mu\left(\mathbf{x}\right), \mathcal{L}_{\mathbf{x}}\kappa\left(\mathbf{x}, \mathbf{x}'\right)\mathcal{L}_{\mathbf{x}'}^{\mathsf{T}} : \mathbb{R}^n \times \mathbb{R}^n \to \mathbb{R}^{a \times a}\right), \tag{8}$$

where $\mathcal{L}_{\mathbf{x}}$ and $\mathcal{L}_{\mathbf{x}'}$ denote the operation of $\mathcal{L}$ on the first and second argument of $\kappa\left(\mathbf{x}, \mathbf{x}'\right)$, respectively (Jidling et al., 2017; Agrell, 2019; Lange-Hegermann, 2021; Besginow & Lange-Hegermann, 2022). To make $\mathcal{L}_{\mathbf{x}}\kappa\left(\mathbf{x}, \mathbf{x}'\right)\mathcal{L}_{\mathbf{x}'}^{\mathsf{T}}$ a valid covariance function, its underlying kernel $\kappa\left(\mathbf{x}, \mathbf{x}'\right)$ needs to be twice differentiable in $\mathbb{R}^n$. See Appendix A.2 for further details on linear operations on GPs.

## 3.1 CURL-FREE KERNEL

The gradient operator ($\nabla := \left[\partial_{x_1}, \ldots, \partial_{x_n}\right]^{\mathsf{T}} \in R^{n \times 1}$) defines a surjective mapping from the space of smooth scalar fields to the space of curl-free vector fields (Do Carmo, 1998), so $\mathbf{f}_{curl}$ can be represented by $\mathbf{f}_{curl} = \nabla V$, where $V \in C^{\infty}\left(\mathbb{R}^n, \mathbb{R}\right)$ is called the scalar potential of $\mathbf{f}_{curl}$. Since the gradient operation defines a linear transformation, if a GP with a scalar kernel $\kappa_V$ is assumed on $V$, the distribution of $\mathbf{f}_{curl}$ is again a GP. According to Eq. 8, the curl-free GP over $\mathbf{f}_{curl}$ is given by

$$\mathbf{f}_{curl} \sim \mathcal{GP}\left(\mathbf{0}, \kappa_{curl} = \nabla_{\mathbf{x}}\kappa_V\left(\mathbf{x}, \mathbf{x}'\right)\nabla_{\mathbf{x}'}^{\mathsf{T}}\right), \tag{9}$$

where $\kappa_{curl} : \mathbb{R}^n \times \mathbb{R}^n \to \mathbb{R}^{n \times n}$ is a matrix-valued kernel constructed by the Hessian of the scalar kernel $\kappa_V$, consisting of all second-order partial derivatives in $\mathbf{x}$ and $\mathbf{x}'$, with the entry in the $i$-th row and $j$-th column given by

$$\left[\kappa_{curl}\left(\mathbf{x}, \mathbf{x}'\right)\right]_{i,j} = \mathrm{Cov}\left[\partial V\left(\mathbf{x}\right)/\partial x_i, \partial V\left(\mathbf{x}'\right)/\partial x_j'\right] = \partial^2 \kappa_V\left(\mathbf{x}, \mathbf{x}'\right)/\partial x_i \partial x_j'. \tag{10}$$

By this construction, if $\kappa_V$ induces a GP with realizations dense in $C^{\infty}\left(\mathbb{R}^n, \mathbb{R}\right)$, the set of realizations of $\mathcal{GP}\left(\mathbf{0}, \kappa_{curl}\right)$ is dense in the space of curl-free vector fields, because a surjective mapping maps dense sets to dense sets.

## 3.2 DIVERGENCE-FREE KERNEL

A div-free vector field can be constructed from a skew-symmetric matrix field (Barbarosie, 2011; Kelliher, 2021; Richter-Powell et al., 2022). Specifically, let $\mathbf{A} : \mathbb{R}^n \to \mathbb{R}^{n \times n}$ be a skew-symmetric matrix-valued function, then a div-free vector field $\mathbf{f}_{div}$ can be represented by taking row-wise divergence of $\mathbf{A}$, i.e.,

$$\mathbf{f}_{div} = \left[\nabla \cdot \mathbf{A}_1, \ldots, \nabla \cdot \mathbf{A}_n\right]^{\mathsf{T}}, \tag{11}$$

where $\mathbf{A}_i : \mathbb{R}^n \to \mathbb{R}^n$ is the $i$-th row of $\mathbf{A}$. The skew-symmetric matrix field $\mathbf{A}$ of size $n \times n$ can be compactly represented by $m = n\left(n - 1\right)/2^2$ scalar functions $u_{ij} \in C^{\infty}\left(\mathbb{R}^n, \mathbb{R}\right)$:

$$\mathbf{A} = \begin{bmatrix} 0 & u_{12} & \ldots & u_{1n} \\ -u_{12} & 0 & \ldots & u_{2n} \\ \vdots & \vdots & \ddots & \vdots \\ -u_{1n} & -u_{2n} & \ldots & 0 \end{bmatrix} = \sum_{i=1}^{n-1}\sum_{j=i+1}^{n}\Phi_{ij}u_{ij}, \tag{12}$$

where $\Phi_{ij} \in \mathbb{R}^{n \times n}$ is a matrix with its $(i, j)$-th entry equal to 1, $(j, i)$-th entry equal to -1, and all other entries equal to 0. Then, the div-free vector field given by Eq. 11 can be reformulated as a linear transformation:

$$\mathbf{f}_{div}\left(\mathbf{x}\right) = \sum_{i=1}^{n-1}\sum_{j=i+1}^{n}\psi_{ij}u_{ij}\left(\mathbf{x}\right) = \Psi\mathbf{u}\left(\mathbf{x}\right), \tag{13}$$

where $\psi_{ij} = \Phi_{ij}\nabla \in R^{n \times 1}$ is a column vector obtained by linearly transforming the gradient operator. The $m$ column vectors $\psi_{ij}$ can be aggregated in a matrix $\Psi \in R^{n \times m}$, and the $m$ corresponding scalar functions $u_{ij}$ are collected in a vector-valued function $\mathbf{u} : \mathbb{R}^n \to \mathbb{R}^m$. $\Psi\left[\cdot\right]$ is a matrix of linear differential operators, so to use a GP to model $\mathbf{f}_{div}$, we can proceed by assuming a GP prior over $\mathbf{u} \sim \mathcal{GP}\left(\mathbf{0}, \kappa_{\mathbf{u}} : \mathbb{R}^n \times \mathbb{R}^n \to \mathbb{R}^{m \times m}\right)$, then based on the closure of GPs under linear transformation (Eq. 8), the GP prior over $\mathbf{f}_{div}$ (Eq. 13) can be constructed by

$$\mathbf{f}_{div} \sim \mathcal{GP}\left(\mathbf{0}, \Psi_{\mathbf{x}}\kappa_{\mathbf{u}}\left(\mathbf{x}, \mathbf{x}'\right)\Psi_{\mathbf{x}'}^{\mathsf{T}}\right), \tag{14}$$

---

[2]$m$ is the number of entries above the diagonal. Each off-diagonal element of the matrix corresponds to a scalar function, with elements below the main diagonal as the negatives of those above.

where $\kappa_{\mathbf{u}}$ is a scalar-valued kernel for two dimensional systems ($n = 2$, $m = 1$), and is a matrix-valued kernel for $n > 2$. The GP model given by Eq. 14 can be used to approximate arbitrary div-free vector fields, because the representation (Eq. 11) has been shown to be maximally expressive (*i.e.* universal) by Richter-Powell et al. (2022).

### 3.3 IDENTIFIABILITY AND CONSTRAINTS

With the curl-free and div-free kernels, our objective is to learn physically interpretable representations of a dynamical system based on the HHD-GP model. However, the HHD is always not unique due to the existence of harmonic components $\mathbf{f}_{harm}$ (vector fields satisfying both $\nabla \wedge \mathbf{f}_{harm} = \mathbf{0}$ and $\nabla \cdot \mathbf{f}_{harm} = 0$, *e.g.*, constant vector fields). For the HHD of a dynamical system with the true functional decomposition $\mathbf{f}_{curl}^*$ and $\mathbf{f}_{div}^*$,

$$\mathbf{f} = (\mathbf{f}_{curl}^* + \mathbf{f}_{harm}) + (\mathbf{f}_{div}^* - \mathbf{f}_{harm}) \tag{15}$$

is a valid HHD for arbitrary $\mathbf{f}_{harm}$, which thus makes the HHD-GP model non-identifiable, meaning that from the same training data, we may learn different decompositions giving the same predictions. This is not desirable because we expect the learned dynamical model to be interpretable: the curl-free and div-free components $\mathbf{f}_{curl}$, $\mathbf{f}_{div}$ are physically meaningful.

To mitigate the identifiability problem in additive regression models, an effective method is to impose constraints on their component models (Durrande et al., 2012; 2013; Martens, 2019; Lu et al., 2022). The imposed constraints can affect the decomposition results of the additive models. Therefore, to ensure that HHD-GP can produce a scientific decomposition, we desire constraints that respect the inherent characteristics of dynamical systems. And, as another primary goal, incorporating prior knowledge of a system into a GP model can also improve its prediction accuracy and learning efficiency. Therefore, in the next section, we present how to impose symmetry-based constraints on the curl-free and div-free GP models.

## 4 SYMMETRY CONSTRAINTS

### 4.1 EQUIVARIANCE AND INVARIANCE

Symmetry is a fundamental geometric property prevalent in dynamical systems in natural (Livio, 2012), and is usually described by the concept of equivariance and invariance:

**Definition 4.1** (Equivariance and Invariance). *Let $\mathcal{G}$ be a group acting on $\mathbb{R}^n$ through a smooth map $L : \mathcal{G} \times \mathbb{R}^n \to \mathbb{R}^n$. The dynamical system $\mathbf{f} : \mathbb{R}^n \to \mathbb{R}^n$ is said to be $\mathcal{G}$-equivariant if*

$$(\mathbf{f} \circ L_g)(\mathbf{x}) = \mathbf{J}_{L_g}(\mathbf{x}) \mathbf{f}(\mathbf{x}), \forall \mathbf{x} \in \mathbb{R}^n, g \in \mathcal{G}, \tag{16}$$

*where $L_g(\mathbf{x}) := L(g, \mathbf{x})$, and $\mathbf{J}_{L_g}$ denotes the Jacobian matrix of $L_g$. Then, $\mathcal{G}$ is termed the symmetry group of the dynamical system. In particular, if $\mathbf{J}_{L_g}$ is the identity matrix (i.e., $\mathbf{f} \circ L_g = \mathbf{f}, \forall g \in \mathcal{G}$), the dynamical system $\mathbf{f}$ is said to be $\mathcal{G}$-invariant.*

From the equivariance condition (Eq. 16) of the vector field, it follows the system's trajectory commutes with the action map. For vector fields on $\mathbb{R}^n$, the symmetry group $\mathcal{G}$ is commonly a subgroup of the Euclidean group $E(n)$, which comprises all intuitive geometric transformations in $\mathbb{R}^n$ (see Appendix A.3 for a brief introduction). The symmetry constraints refer to that we expect the learned curl-free and div-free vector fields to be $\mathcal{G}$-equivariant. With the representation of their GP models, we demonstrate that the symmetries can be enforced via the design of suitable kernel functions.

### 4.2 SYMMETRY-PRESERVING CURL-FREE GP

The curl-free GP (Eq. 9) is constructed by transforming another GP over a potential function, implying that we can impose constraints of symmetry on the curl-free GP by designing a suitable potential GP. Therefore, we start by exploring how to construct potential functions to obtain curl-free vector fields with the desired equivariance. As expected, the following theorem holds:

**Theorem 4.1.** *Let $\mathcal{G}$ be a Euclidean group or its subgroup, and let $V : \mathbb{R}^n \to \mathbb{R}$ be a $\mathcal{G}$-invariant scalar function. Then, the curl-free vector field $\mathbf{f}_{curl} : \mathbb{R}^n \to \mathbb{R}^n$ defined by $\mathbf{f}_{curl}(\mathbf{x}) = \nabla V(\mathbf{x})$ is $\mathcal{G}$-equivariant.*

See Appendix A.4.1 for the proof. Theorem 4.1 shows that a $\mathcal{G}$-invariant scalar potential $V$ can yield a $\mathcal{G}$-equivariant gradient field, indicating that if any realization $V$ of $\mathcal{GP}(0, \kappa_V)$ is constrained to

be $\mathcal{G}$-invariant, then its pushforward GP over $\nabla V \sim \mathcal{GP}\left(\mathbf{0}, \nabla_{\mathbf{x}}\kappa_V \nabla_{\mathbf{x}'}^{\mathsf{T}}\right)$ can induce the space of $\mathcal{G}$-equivariant curl-free vector fields.

It is obvious that a $\mathcal{G}$-invariant scalar potential $V$ can be constructed by integrating some non-invariant function $h : \mathbb{R}^n \to \mathbb{R}$ over the symmetry group: $V = \int_{\mathcal{G}} \left(h \circ L_g\right) dg$, where the measure $dg$ is called *Haar measure*, which exists for locally compact topological groups and finite groups. Therefore, by assuming that $h$ is distributed as $h \sim \mathcal{GP}\left(0, \kappa_h\right)$, we can construct the GP prior over the $\mathcal{G}$-invariant scalar potential as $V \sim \mathcal{GP}\left(0, \kappa_V\right)$, with its kernel $\kappa_V$ given by

$$\kappa_V = \text{Cov}\left[\int_{\mathcal{G}} h\left(L_g\left(\mathbf{x}\right)\right) dg, \int_{\mathcal{G}} h\left(L_g\left(\mathbf{x}'\right)\right) dg\right] = \int_{\mathcal{G}}\int_{\mathcal{G}} \kappa_h\left(L_g\left(\mathbf{x}\right), L_{g'}\left(\mathbf{x}'\right)\right) dg dg'. \quad (17)$$

This kernel is called the *Haar-integration kernel* (Haasdonk et al., 2005). While it provides a general method for constructing kernels for $\mathcal{G}$-invariant functions, the double integral can be computationally expensive. If the kernel $\kappa_h$ is invariant to any $g \in \mathcal{G}$ in the sense that $\kappa\left(\mathbf{x}, \mathbf{x}'\right) = \kappa\left(L_g\left(\mathbf{x}\right), L_g\left(\mathbf{x}'\right)\right)^3$, a complexity reduction of Eq. 17 by one square-root can be performed by

$$\int_{\mathcal{G}}\int_{\mathcal{G}} \kappa_h\left(L_g\left(\mathbf{x}\right), L_{g'}\left(\mathbf{x}'\right)\right) dg dg' = \int_{\mathcal{G}}\int_{\mathcal{G}} \kappa_h\left(\mathbf{x}, L_{g^{-1}g'}\left(\mathbf{x}'\right)\right) dg dg' = |\mathcal{G}| \int_{\mathcal{G}} \kappa_h\left(\mathbf{x}, L_g\left(\mathbf{x}'\right)\right) dg,$$

$$(18)$$

where $|\mathcal{G}| = \int_{\mathcal{G}} dg$, and it denotes the cardinality of $\mathcal{G}$ when the group is finite.

### 4.3 Symmetry-preserving Divergence-free GP

To incorporate the equivariance condition (Eq. 16) into realizations of the div-free GP (Eq. 14), we construct the skew-symmetric matrix field $\mathbf{A}$ from a vector-valued function. Specifically, given a smooth vector field $\mathbf{v} \in C^\infty\left(\mathbb{R}^n, \mathbb{R}^n\right)$, $\mathbf{A}$ is constructed by $\mathbf{A} = \mathbf{J_v} - \mathbf{J_v^{\mathsf{T}}}$, where $\mathbf{J_v}$ denotes the Jacobian of $\mathbf{v}$ with its $(i, j)$-th entry given by $\partial v_i / \partial x_j$. Then the component function $u_{ij}$ in Eq. 12 is given by $u_{ij} = \partial v_i / \partial x_j - \partial v_j / \partial x_i$. By this construction, the symmetry of the div-free vector field $\mathbf{f}_{div}$ is governed by the symmetry of its vector potential $\mathbf{v}$. In particular, a $\mathcal{G}$-equivariant $\mathbf{v}$ can produce a $\mathcal{G}$-equivariant $\mathbf{f}_{div}$, and it is formalized in the following theorem:

**Theorem 4.2.** *Let $\mathcal{G}$ be a Euclidean group or its subgroup, and let $\mathbf{v} : \mathbb{R}^n \to \mathbb{R}^n$ be a $\mathcal{G}$-equivariant vector field. Then the divergence-free vector field $\mathbf{f}_{div} = \left[\nabla \cdot \mathbf{A}_1, \dots, \nabla \cdot \mathbf{A}_n\right]^{\mathsf{T}}$ is $\mathcal{G}$-equivariant, where $\mathbf{A}_i$ denotes the $i$-th row of the skew-symmetric matrix-valued function $\mathbf{A} = \mathbf{J_v} - \mathbf{J_v^{\mathsf{T}}}$.*

The proof can be found in Appendix A.4.2. By this theorem, we then proceed by assuming a GP prior over the vector potential $\mathbf{v} \sim \mathcal{GP}\left(\mathbf{0}, \kappa_{\mathbf{v}}\right)$, and to constrain $\mathbf{v}$ to be $\mathcal{G}$-equivariant, we build its kernel $\kappa_{\mathbf{v}} \in \mathbb{R}^{n \times n}$ in the form of the *Group Integration Matrix kernel* (GIM-kernel) (Reisert & Burkhardt, 2007; Reisert, 2008), which is constructed by:

$$\kappa_{\mathbf{v}}\left(\mathbf{x}, \mathbf{x}'\right) = \int_{\mathcal{G}} \kappa\left(\mathbf{x}, L_g\left(\mathbf{x}'\right)\right) \mathbf{J}_{L_g} dg, \quad (19)$$

where $\kappa$ is some arbitrary scalar-valued kernel satisfying $\kappa\left(\mathbf{x}, \mathbf{x}'\right) = \kappa\left(L_g\left(\mathbf{x}\right), L_g\left(\mathbf{x}'\right)\right)$ for all $g \in \mathcal{G}$. The GIM-kernel spans a Reproducing Kernel Hilbert Space (RKHS) of functions with the desired equivariance (Reisert, 2008). So we can then use $\kappa_{\mathbf{v}}$ (Eq. 19) to construct the GP prior over $\mathbf{u}$ in Eq. 13, where the covariance between components $u_{ij}$ and $u_{kq}$ is given by

$$\left[\kappa_{\mathbf{u}}\right]_{ij,kq} = \text{Cov}\left[u_{ij} = \frac{\partial v_i}{\partial x_j} - \frac{\partial v_j}{\partial x_i}, u_{kq} = \frac{\partial v_k}{\partial x_q} - \frac{\partial v_q}{\partial x_k}\right]$$

$$= \frac{\partial^2}{\partial x_j \partial x'_q}\left[\kappa_{\mathbf{v}}\right]_{i,k} + \frac{\partial^2}{\partial x_i \partial x'_k}\left[\kappa_{\mathbf{v}}\right]_{j,q} - \frac{\partial^2}{\partial x_j \partial x'_k}\left[\kappa_{\mathbf{v}}\right]_{i,q} - \frac{\partial^2}{\partial x_i \partial x'_q}\left[\kappa_{\mathbf{v}}\right]_{j,k}. \quad (20)$$

Finally, this matrix-valued kernel $\kappa_{\mathbf{u}}$ is transformed by Eq. 14 to construct the div-free GP, of which the realizations are guaranteed to be $\mathcal{G}$-equivariant div-free vector fields, according to Theorem 4.2.

## 5 Related work

**Learning with div/curl-free constraints** Div-free vector fields are a focal point of mathematical physics and have been well exploited by machine learning models for learning conservative dynamics, with the most well-known examples being neural networks (NNs) (Greydanus et al., 2019;

---

[3]For $\mathcal{G} \subseteq E\left(n\right)$, it holds that $\|L_g\left(\mathbf{x}\right) - L_g\left(\mathbf{x}'\right)\| = \|\mathbf{x} - \mathbf{x}'\|$, for all $\mathbf{x}, \mathbf{x}' \in \mathbb{R}^n$, and $g \in \mathcal{G}$. Therefore, $\kappa\left(\mathbf{x}, \mathbf{x}'\right) = \kappa\left(L_g\left(\mathbf{x}\right), L_g\left(\mathbf{x}'\right)\right)$ is satisfied if $\kappa$ is an isotropic kernel, *i.e.*, $\kappa\left(\mathbf{x}, \mathbf{x}'\right) = \kappa\left(\|\mathbf{x} - \mathbf{x}'\|\right)$, common examples of which are the squared exponential kernel and the Matérn class of kernels (cf. chap.4 in Rasmussen & Williams (2005)).

Toth et al., 2019) and GPs (Rath et al., 2021; Ross & Heinonen, 2023) for learning Hamiltonian dynamics. All Hamiltonian vector fields are div-free, but not vice versa. To learn more general div-free vector fields as solutions of the continuity equation, Richter-Powell et al. (2022) introduced an NN architecture to parameterize a universal representation of div-free vector fields. Based on the same representation, we constructed div-free kernels for GPs from matrix-valued kernels, which is an extension of the method of constructing div-free kernels by the curl operator in $\mathbb{R}^3$ (Narcowich & Ward, 1994; Lowitzsch, 2002; Wendland, 2009). According to Maxwell's equations, div-free kernels were combined with curl-free kernels by Wahlström et al. (2013); Wahlström (2015) to model magnetic fields. They assumed direct access to noisy observations of the div-free and curl-free components separately, whereas in this work, our goal is to recover the individual components from noisy observations of their sum. Similar prediction problem was studied by Berlinghieri et al. (2023), they combined 2D div-free and curl-free GP priors in the form of HHD to reconstruct planar ocean current fields, and recovered their divergence. Their model has the same formulation as ours when the dimension of HHD-GP is two, but we further developed a symmetry-preserving extension of HHD-GP to solve its non-identifiability problem. Another similar work is Dissipative Hamiltonian neural network (D-HNN) (Greydanus & Sosanya, 2022), which compensated HNN (Greydanus et al., 2019) with a curl-free part to model both conservative and dissipative dynamics simultaneously, but D-HNN is not applicable to odd-dimensional system due to its construction.

**Learning with symmetry** Symmetries are another important aspect of priors that can be incorporated into machine learning models. Motivated by the success of the translation-invariant NNs (LeCun et al., 1989), network architectures with symmetries to more general transformations have been proposed, such as steerable CNNs (Weiler & Cesa, 2019; Cesa et al., 2021) and graph NNs (Maron et al., 2019; Satorras et al., 2021) equivariant to Euclidean symmetries. They achieved great success in improving generalization of the models. With the same motivation, kernel methods incorporating symmetries have also been developed. To make predictions invariant to transformations of inputs, Haasdonk et al. (2005) constructed kernels using Haar integration. And based on similar integration technique, Reisert & Burkhardt (2007) developed the Group Integration Matrix Kernel (GIM-kernel) to learn equivariant functions, which was later used by Ridderbusch et al. (2021) to learn dynamical systems with symmetries. In this work, we constructed GP models to impose Euclidean symmetries to div/curl-free vector fields, which to the best of our knowledge has not been explored by the machine learning community.

## 6 EXPERIMENTS AND RESULTS

**Data and tasks** We evaluated our method on three classical physical systems: a damped mass-spring system, a damped pendulum, and a Chua circuit. The governing equations of these systems and their symmetries are detailed in Appendix A.5. We generated the training data $\{(\mathbf{x}, \dot{\mathbf{x}})\}$ by uniformly sampling states $\mathbf{x}$ in their phase space, and each of their derivative observations $\dot{\mathbf{x}}$ is corrupted by an additive Gaussian noise with a standard deviation of $0.01$. And for each system, we sampled 20 pairs of data for training. We first evaluated the models' performance in terms of learning ODEs. Specifically, the first evaluation metric focuses on the accuracy of the models in predicting state derivatives, as measured by the *root mean squared error* (RMSE), $(\frac{1}{m} \sum_{i=1}^{m} \|\hat{\mathbf{x}}_i - \dot{\mathbf{x}}_i\|^2)^{\frac{1}{2}}$, where $\hat{\mathbf{x}}_i$ and $\dot{\mathbf{x}}_i$ are the predicted and true state derivatives, respectively, and $m$ is the number of test data. The lower this metric, the better. And the test set $\{(\mathbf{x}_i, \dot{\mathbf{x}}_i)\}_{i=1}^{m}$ was generated by sampling a grid with a resolution of 10 points along each dimension of the dynamical systems. Another evaluation metric for learning ODEs focuses on the accuracy in predicting state trajectories over time, as measured by the *valid prediction time* (VPT), $\frac{1}{T} \arg \min_t \{\text{NRMSE} (\hat{\mathbf{x}}_t, \mathbf{x}_t) > \epsilon, \forall\, 0 \leq t \leq T\}$. VPT calculates the first time step $t$ at which the *normalized root mean square error* (NRMSE) between the predicted state $\hat{\mathbf{x}}_t$ and the ground truth $\mathbf{x}_t$ exceeds a given threshold, and $\text{NRMSE} (\hat{\mathbf{x}}_t, \mathbf{x}_t) = ((\hat{\mathbf{x}}_t - \mathbf{x}_t)^{\mathsf{T}} \Sigma (\hat{\mathbf{x}}_t - \mathbf{x}_t) / n)^{\frac{1}{2}}$, $\Sigma = \text{diag} (1/\sigma_1, \ldots, 1/\sigma_n)$, with $\sigma_i$ denoting the variance of the $i$-th dimension of the true trajectory. The VPT measures how long the predicted trajectory remains close to the true trajectory, so the higher this indicator, the better. In our experiments, the value of $\epsilon$ is set to be $0.01$. To alleviate the dependency on the initial condition, we reported the VPT averaged over trajectories simulated from 50 randomly sampled initial conditions. And a trajectory from an initial condition was solved by the Dormand–Prince method (dopri5) (Dormand & Prince, 1986) implemented in *torchdiffeq*[4], integrating forward in time at a frequency of 25 Hz for 15 seconds, with

---

[4] https://github.com/rtqichen/torchdiffeq

the relative and absolute tolerances of $10^{-6}$. One benefit of our model is that it can decompose the dynamics into its div-free and curl-free components, thus providing new insights to the dynamical systems. To show this, we evaluated our model by another task: we predicted the energy evolution along the trajectories of the systems.

**Baselines and implementation details**  We compared our models, HHD-GP and its symmetry-preserving extension SPHHD-GP, with Dissipative Hamiltonian neural network (D-HNN) (Greydanus & Sosanya, 2022), GPs that only involve div-free kernels for learning conservative dynamics (Div-GP) (Rath et al., 2021; Ross & Heinonen, 2023), and GPs with Group Integration Matrix Kernels (GIM-GP) (Reisert & Burkhardt, 2007) that can incorporate symmetries. Another baseline is GPs with independent kernels (Ind-GP), which model each dimension of a dynamical system with an independent scalar GP. Due to its easy implementation, Ind-GP is widely used in modeling robotic systems (Deisenroth et al., 2015; Kamthe & Deisenroth, 2018) and magnetic fields (Vallivaara et al., 2010; 2011). See Appendix A.7 for the implementation details of these models.

## 6.1 COMPARISON OF LEARNING ODE MODELS

We first compare the models in predicting state derivatives and trajectories, measured by the RMSEs and VPTs in Table 1, respectively. The performance of Div-GP is limited because it can only model conservative dynamics. HHD-GP improves its performance by compensating with a curl-free kernel, which offsets the strong inductive bias imposed by the div-free kernel. And the performance of HHD-GP is better than that of another HHD-based model, D-HNN, because the low data efficiency of NNs makes it hard for D-HNN to capture dynamics using noisy and sparse training data, so actually the performance of D-HNN is worse than either of the GP methods. Moreover, D-HNN is not applicable to the Chua circuit because its phase space is odd-dimensional. As another model without inductive bias, Ind-GP performs similarly to HHD-GP in the damped mass-spring system. However, in the two remaining systems with more complex behavior, HHD-GP performs better than Ind-GP, because the kernel of Ind-GP fails to model correlations between different dimensions of a dynamical system. Then, by incorporating symmetry priors into GPs, GIM-GP performs better than the above models but not as well as SPHHD-GP, because learning in the form of HHD allows SPHHD-GP to exploit more implicit symmetries in the dynamical systems. SPHHD-GP performs the best in all systems. Appendix A.8.2 contains plots of the trajectory predictions for each system. And in Appendix A.8.3 we further investigated the effect of noise and amount of training data on the models performance, and the results show that our model (SPHHD-GP) is more robust to noise level and data amount relative to the baselines.

Table 1: Comparison of our models to baselines. The RMSE and the VPT are recorded in the scale of $\times 10^{-2}$ and in the form of mean $\pm$ standard deviation. Bold font indicates best results. All of the experimental results are based on 10 independent experiments performed by resampling the training sets and model initial parameters.

| Model | Damped Mass Spring | | Damped Pendulum | | Chua Circuit | |
|---|---|---|---|---|---|---|
| | RMSE ↓ | VPT ↑ | RMSE ↓ | VPT ↑ | RMSE ↓ | VPT ↑ |
| D-HNN | $34.42 \pm 8.37$ | $1.36 \pm 0.51$ | $186.90 \pm 27.19$ | $0.40 \pm 0.07$ | N/A | N/A |
| Div-GP | $20.70 \pm 38.36$ | $1.07 \pm 0.28$ | $57.59 \pm 22.65$ | $1.21 \pm 0.23$ | $141.44 \pm 63.04$ | $0.37 \pm 0.09$ |
| Ind-GP | $0.93 \pm 0.35$ | $30.70 \pm 28.77$ | $76.45 \pm 27.85$ | $3.14 \pm 1.18$ | $35.16 \pm 14.72$ | $1.70 \pm 0.43$ |
| GIM-GP | $0.44 \pm 0.21$ | $39.95 \pm 25.05$ | $17.34 \pm 8.05$ | $11.22 \pm 5.20$ | $8.55 \pm 4.36$ | $2.30 \pm 0.57$ |
| HHD-GP (ours) | $0.87 \pm 0.36$ | $33.81 \pm 27.52$ | $24.87 \pm 16.75$ | $4.66 \pm 1.16$ | $19.97 \pm 6.98$ | $1.29 \pm 0.17$ |
| SPHHD-GP (ours) | $\mathbf{0.31 \pm 0.19}$ | $\mathbf{44.93 \pm 24.09}$ | $\mathbf{8.21 \pm 6.24}$ | $\mathbf{22.67 \pm 16.09}$ | $\mathbf{2.25 \pm 0.69}$ | $\mathbf{5.44 \pm 1.05}$ |

## 6.2 COMPARISON OF PREDICTING ENERGY EVOLUTION

According to the connection between HHD and the generalized Hamiltonian formalism, the div-free component in HHD is closely related to the energy of a dynamical system, so the HHD-based models including HHD-GP, SPHHD-GP and D-HNN can be used to predict energy evolution of the systems in the experiments (see Appendix A.7.7 for implementation details). The results are shown in Fig. 2, where we can find that D-HNN and HHD-GP fail to provide physically plausible results and their predictions have large deviations from the ground truth, along with significant variances. In contrast, predictions of SPHHD-GP are highly accurate and closely aligned with the true values. One reason is that the symmetry priors used by SPHHD-GP improves the generalization performance of the model, but more importantly, the priors solve the problem of non-identifiability suffered by HHD-GP and D-HNN. A numerial comparision of energy prediction is given in Appendix A.8.1.

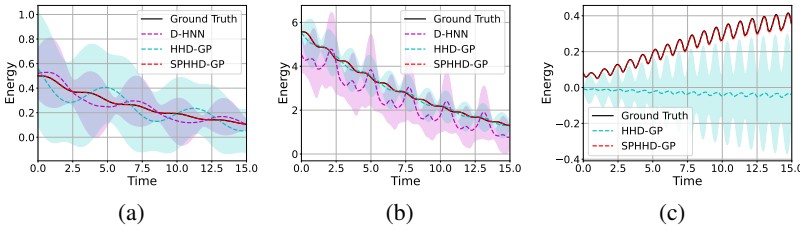

Figure 2: Energy prediction $\hat{H}(\mathbf{x}_t)$ along the trajectory initialized at (a) $(1.0, 0.0)$ of the damped mass-spring system; (b) $(1.5, 0.0)$ of the damped pendulum; (c) $(0.05, -0.4, -0.05)$ of the Chua circuit, where $\hat{H}(\cdot)$ is the estimated Hamiltonian function and $\{\mathbf{x}_t\}$ is a true trajectory of the system.

To further illustrate the non-identifiability problem, we present the results of the mass-spring system when the number of training data is increasing in Fig. 3. As expected, the RMSE of derivative predictions decreases as the number of training data increases. However, the corresponding RMSE of energy prediction of HHD-GP and D-HNN cannot converge. Fig. 4 presents a learned decomposition when the number of training data is $180$. Although HHD-

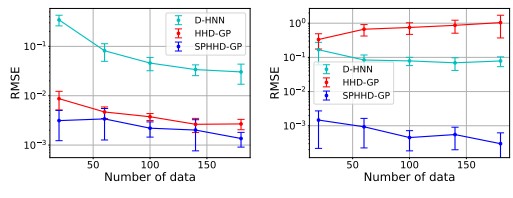

(a) RMSE of derivatives    (b) RMSE of energy

Figure 3: Results with increasing number of data.

GP and SPHHD-GP capture highly similar system dynamics (the first column in Fig. 4a), they learn completely different decompositions (the second and third columns in Fig. 4a). Compared with the ground truth in Fig. 1, SPHHD-GP learns the physically correct decomposition, so it can accurately predict the system energy. From Fig. 4b we can observe that the predictions of HHD-GP have large variance, meaning that the model is less certain in isolating individual effects from other terms. In Appendix A.6, we provide a theoretical verification that for the three dynamical systems in our experiments the non-uniqueness of HHD is solved through forced symmetries.

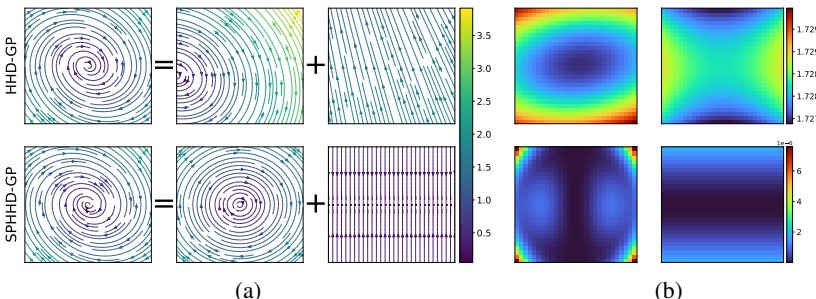

Figure 4: Example predicted HHD of the damped-mass spring system by HHD-GP (the first row) and SPHHD-GP (the second row). (a): predicted vector field (the first column) with its div-free (the second column) and curl-free (the third column) components; (b): the associated variance of the div-free (the first column) and curl-free (the second column) predictions.

## 7    CONCLUSION AND FUTURE WORK

Our work develops an additive GP model whose component is either free of divergence or of curl, the two most ubiquitous differential invariants of vector fields in natural, and we constrain the div/curl-free kernels to preserve symmetries of the underlying system. These symmetry-preserving kernels not only improve the accuracy of predictions but also make the model identifiable, thus the energy evolution of a dynamical system can be predicted. A limitation of our model is that its computational complexity grows cubically with the amount of training data and the dimension of the dynamical system, so a future research direction is to combine the proposed model with methods such as sparse variational inference and low-rank approximation to reduce the computational complexity. Another future direction is to extend our model to exploit the connection of HHD with more dynamical systems. For example, there are recent advances in using HHD to construct Lyapunov functions (Suda, 2019) and to reformulate the Navier-Stokes equations (Caltagirone, 2021). So our model have potential to achieve good performance in learning stable dynamics and fluid dynamics.

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

# A   APPENDIX

## A.1   HHD AND GENERALIZED HAMILTONIAN FORMALISM

This section provides a brief introduction to the connections between the Helmholtz-Hodge decomposition (HHD) and the generalized Hamiltonian formalism necessary to understand this work.

We begin with a brief review of Hamiltonian mechanics (Arnol'd, 2013). For a dynamical system with $N$ degrees of freedom, the Hamiltonian formalism describes the system by defining a scalar function $H(\mathbf{x})$ known as the *Hamiltonian*, where the system state $\mathbf{x} = (\mathbf{q}, \mathbf{p}) \in \mathbb{R}^{2N}$ is described by generalized coordinates $\mathbf{q} \in \mathbb{R}^N$ and $\mathbf{p} \in \mathbb{R}^N$ in the phase space, corresponding to generalized position and momentum, respectively. The time evolution of $(\mathbf{q}, \mathbf{p})$ is governed by the symplectic gradient of its Hamiltonian, *i.e.*, $\dot{\mathbf{q}} = \frac{\partial H}{\partial \mathbf{p}}$, $\dot{\mathbf{p}} = -\frac{\partial H}{\partial \mathbf{q}}$. If the Hamiltonian is not explicitly time-dependent, $[\dot{\mathbf{q}}, \dot{\mathbf{p}}]^\mathsf{T}$ defines a divergence-free vector field ($\frac{\partial}{\partial \mathbf{q}}\frac{\partial H}{\partial \mathbf{p}} - \frac{\partial}{\partial \mathbf{p}}\frac{\partial H}{\partial \mathbf{q}} = 0$) whose flows conserve the Hamiltonian. By interpreting the Hamiltonian as energy, the Hamiltonian vector field can model systems with energy conservation. However, real-life systems often suffer from energy dissipation. In physical systems governed by an autonomous ODE, any energy variation occurs along a volume change in phase space and vice versa. Therefore, to account for energy dissipation, a term responsible for the volume contraction of the phase space can be introduced into the Hamiltonian dynamics. This term represents the energy lost by the system due to various dissipative forces and is typically modeled by gradient fields of some scalar functions $D(\cdot)$, so the motion equations of the dissipative Hamiltonian system can be given by

$$\mathbf{f}(\mathbf{q}, \mathbf{p}) = \underbrace{\left[\frac{\partial H}{\partial \mathbf{p}}, -\frac{\partial H}{\partial \mathbf{q}}\right]^\mathsf{T}}_{\text{divergence}-\text{free}} + \underbrace{\nabla D(\mathbf{q}, \mathbf{p})}_{\text{curl}-\text{free}}. \tag{21}$$

In a dissipative Hamiltonian system, the Hamiltonian still represents the system's total energy, but the additional damping term causes the system to lose energy over time. One common example of the damping term is induced by the *Rayleigh function*, $D(\mathbf{p}) = -\frac{1}{2}\mathbf{p}^\mathsf{T} Q \mathbf{p}$, where $Q \in \mathbb{R}^{N \times N}$ is a symmetric positive-definite matrix called the *Rayleigh dissipation matrix*. The Rayleigh function provides an elegant way to include dissipative forces—such as friction, air resistance, and viscosity—in the context of Hamiltonian mechanics.

The dissipative Hamiltonian system (Eq. 21) is in an explicit form of HHD, because the Hamiltonian vector field is divergence-free and the dissipative field $\nabla D$ is curl-free. However, as its divergence-free part is governed by the Hamiltonian equations, Eq. 21 can only be used to describe a subset of even-dimensional systems. To extend its scope, Sarasola et al. (2004) proposed to approach Hamilton energy for dimensionless dynamical systems by using HHD as follows,

$$\begin{cases} \dot{\mathbf{x}} = \mathbf{f}(\mathbf{x}) = \mathbf{f}_{div}(\mathbf{x}) + \mathbf{f}_{curl}(\mathbf{x}), \ \mathbf{x} \in \mathbb{R}^n; \\ \nabla H^\mathsf{T}\mathbf{f}_{div}(\mathbf{x}) = 0; \ \dot{H} = \nabla H^\mathsf{T}\mathbf{f}_{curl}(\mathbf{x}), \end{cases} \tag{22}$$

where $\mathbf{f}_{div}$ and $\mathbf{f}_{curl}$ are divergence-free and curl-free components of a dynamical system, respectively. According to this criterion (Eq. 22), Hamiltonian functions of chaotic systems were approached to calculate the energy for synchronizing two chaotic systems (Sarasola et al., 2004), to analyze the stability of chaotic systems (Zhou et al., 2021), and to design energy modulation-based controllers (Ma et al., 2017).

## A.2   GPS AND LINEAR OPERATORS

Gaussian processes (GPs) are closed under linear transformations (cf. Lemma 2.2 in (Lange-Hegermann, 2021), Lemma 2.1 in (Härkönen et al., 2022)). Let $\mathcal{L}$ be a linear operator acting on realizations of $g \sim \mathcal{GP}(\mu_g(\mathbf{x}), \kappa_g(\mathbf{x}, \mathbf{x}'))$, then under the assumption that $\mathcal{L}$ commutes with expectation, the mean of $\mathcal{L}g$ is given by

$$\mathbb{E}[\mathcal{L}g(\mathbf{x})] = \mathcal{L}\mathbb{E}[g(\mathbf{x})] = \mathcal{L}\mu_g(\mathbf{x}), \tag{23}$$

and the covariance is such that

$$\text{cov}\left[\mathcal{L}_{\mathbf{x}}g\left(\mathbf{x}\right),\mathcal{L}_{\mathbf{x}'}g\left(\mathbf{x}'\right)\right] = \mathbb{E}\left[\left(\mathcal{L}_{\mathbf{x}}g\left(\mathbf{x}\right) - \mathcal{L}_{\mathbf{x}}\mu\left(\mathbf{x}\right)\right)\left(\mathcal{L}_{\mathbf{x}'}g\left(\mathbf{x}'\right) - \mathcal{L}_{\mathbf{x}'}\mu\left(\mathbf{x}'\right)\right)^{\mathsf{T}}\right] \quad (24)$$

$$= \mathcal{L}_{\mathbf{x}}\mathbb{E}\left[\left(g\left(\mathbf{x}\right) - \mu\left(\mathbf{x}\right)\right)\left(g\left(\mathbf{x}'\right) - \mu\left(\mathbf{x}'\right)\right)^{\mathsf{T}}\right]\mathcal{L}_{\mathbf{x}'}^{\mathsf{T}} \quad (25)$$

$$= \mathcal{L}_{\mathbf{x}}\kappa_g\left(\mathbf{x},\mathbf{x}'\right)\mathcal{L}_{\mathbf{x}'}^{\mathsf{T}}. \quad (26)$$

A common application of this technique is to construct GPs with realizations in the solution set of linear differential equations, assuming that $\mathcal{L}$ is a linear differential operator (Jidling et al., 2017; Lange-Hegermann, 2018; 2021; Besginow & Lange-Hegermann, 2022; Härkönen et al., 2022). And similarly, we make use of the closure of GPs under linear differential operators to construct the curl-free and div-free kernels.

Although the transformed kernel (Eq. 26) has been widely used, its validity as a covariance function is rarely discussed in existing works. In a GP framework, a kernel $\kappa\left(\mathbf{x},\mathbf{x}'\right) : \mathbb{R}^n \times \mathbb{R}^n \to \mathbb{R}^{m \times m}$ is a valid covariance function if it is:

*(i)* Symmetric, *i.e.*, $\kappa\left(\mathbf{x},\mathbf{x}'\right) = \kappa\left(\mathbf{x}',\mathbf{x}\right)^{\mathsf{T}}, \forall \mathbf{x},\mathbf{x}' \in \mathbb{R}^n$, and

*(ii)* Positive semidefinite, *i.e.*, $\sum_{ij}\mathbf{c}_i^{\mathsf{T}}\kappa\left(\mathbf{x}_i,\mathbf{x}_j\right)\mathbf{c}_j \geq 0$, for any finite set $\{\mathbf{x}_i\} \subset \mathbb{R}^n$ and $\{\mathbf{c}_i\} \subset \mathbb{R}^m$.

These two conditions for a kernel can be verified if and only if there exists a feature map $\phi\left(\mathbf{x}\right)$ such that $\kappa\left(\mathbf{x},\mathbf{x}'\right) = \phi\left(\mathbf{x}\right)\phi\left(\mathbf{x}'\right)^{\mathsf{T}}$ (Rasmussen & Williams, 2005). So, if $\kappa_g\left(\mathbf{x},\mathbf{x}'\right)$ is a valid kernel, it holds that

$$\mathcal{L}_{\mathbf{x}}\kappa_g\left(\mathbf{x},\mathbf{x}'\right)\mathcal{L}_{\mathbf{x}'}^{\mathsf{T}} = \mathcal{L}_{\mathbf{x}}\phi\left(\mathbf{x}\right)\phi\left(\mathbf{x}'\right)^{\mathsf{T}}\mathcal{L}_{\mathbf{x}'}^{\mathsf{T}} = \left(\mathcal{L}_{\mathbf{x}}\phi\left(\mathbf{x}\right)\right)\left(\mathcal{L}_{\mathbf{x}'}\phi\left(\mathbf{x}'\right)\right)^{\mathsf{T}}. \quad (27)$$

Therefore, the transformed kernel $\mathcal{L}_{\mathbf{x}}\kappa_g\left(\mathbf{x},\mathbf{x}'\right)\mathcal{L}_{\mathbf{x}'}^{\mathsf{T}}$ is guaranteed to be a valid covariance function provided that its underlying kernel $\kappa_g\left(\mathbf{x},\mathbf{x}'\right)$ is. However, when $\mathcal{L}$ is a differential operator, $\kappa_g$ should be twice differentiable, which is satisfied by most of the standard kernels, such as the squared exponential kernel.

### A.3 BASICS FOR EUCLIDEAN GROUP

This section gives the basic definitions about the *Euclidean group*. In the context of this work, the most important example of a symmetry group is the Euclidean group $E\left(n\right)$ or its subgroups. The set of all elements in $E\left(n\right)$ can be denoted as

$$E\left(n\right) = \left\{\left(A,\mathbf{b}\right) \mid A \in O\left(n\right), \mathbf{b} \in \mathbb{R}^n\right\}, \quad (28)$$

where $O\left(n\right) = \left\{A \in \mathbb{R}^{n \times n} \mid AA^{\mathsf{T}} = I\right\}$ is the *orthogonal group*. Any element $g = \left(A,\mathbf{b}\right) \in E\left(n\right)$ represents a translation followed by an orthogonal transformation, the action of $g$ on a point $\mathbf{x} \in \mathbb{R}^n$ is given by a linear mapping:

$$L_g : \mathbb{R}^n \to \mathbb{R}^n, \ \mathbf{x} \mapsto A\left(\mathbf{x} + \mathbf{b}\right). \quad (29)$$

Therefore, $E\left(n\right)$ comprises all *isometries* of a Euclidean space, *i.e.* for all $\mathbf{x},\mathbf{x}' \in \mathbb{R}^n$ and $g \in E\left(n\right)$, we have

$$\left\|L_g\left(\mathbf{x}\right) - L_g\left(\mathbf{x}'\right)\right\| = \left\|\mathbf{x} - \mathbf{x}'\right\|, \quad (30)$$

where $\left\|\cdot\right\|$ is the Euclidean norm. All intuitive geometric transformations in $\mathbb{R}^n$ can be described by subgroups of $E\left(n\right)$, such as

1. **Translation:** The group of all translations in $\mathbb{R}^n$ is denoted by $\left(\mathbb{R}^n, +\right)$. For any $\mathbf{v} \in \mathbb{R}^n$, A translation is a transformation that moves a point $\mathbf{x} \in \mathbb{R}^n$ by $L_{\mathbf{v}}\left(\mathbf{x}\right) = \mathbf{x} + \mathbf{v}$.

2. **Rotation:** The group of all rotations in $\mathbb{R}^n$ is represented by the set of special orthogonal matrices $SO\left(n\right) = \left\{R \in O\left(n\right) \mid \det R = 1\right\}$, where a rotation matrix $R$ transforms a point by $L_R\left(\mathbf{x}\right) = R\mathbf{x}$.

3. **Reflection:** Reflections in $\mathbb{R}^n$ forms subgroups of the orthogonal group $O\left(n\right)$. Reflections correspond to mirror symmetries. They mirror points across a hyperplane. For a hyperplane with a unit normal vector $\mathbf{n}$, the action of a reflection is defined as $L_{\mathbf{n}}\left(\mathbf{x}\right) = \mathbf{x} - 2(\mathbf{x}\cdot\mathbf{n})\mathbf{n}$.

## A.4 Proofs

This section restates and proves the theorems in Section 4, which give the theoretical foundations that we use to enforce symmetry constraints to our HHD-GP model.

### A.4.1 Proof of Theorem 4.1

**Theorem 4.1.** *Let $\mathcal{G}$ be a Euclidean group or its subgroup, and let $V : \mathbb{R}^n \to \mathbb{R}$ be a $\mathcal{G}$-invariant scalar function. Then, the curl-free vector field $\mathbf{f}_{curl} : \mathbb{R}^n \to \mathbb{R}^n$ defined by $\mathbf{f}_{curl}(\mathbf{x}) = \nabla V(\mathbf{x})$ is $\mathcal{G}$-equivariant.*

*Proof.* To prove that $\nabla V(\mathbf{x})$ is $\mathcal{G}$-equivariant, by Definition 4.1 we need to show that for any $\mathbf{x} \in \mathbb{R}^n$ and $g \in \mathcal{G}$, we have

$$\nabla V \circ L_g = \mathbf{J}_{L_g} \nabla V, \tag{31}$$

where $L_g(\mathbf{x}) : \mathbb{R}^n \to \mathbb{R}^n$ is the action of $g$ on $\mathbf{x}$, and $\mathbf{J}_{L_g}$ is the Jacobian matrix of $L_g$. To show this, we first recall the chain rule of the gradient operator,

$$\nabla(V \circ L_g) = \mathbf{J}_{L_g}^\mathsf{T}(\nabla V \circ L_g), \tag{32}$$

where $\mathbf{J}_{L_g}^\mathsf{T}$ denotes the transpose of $\mathbf{J}_{L_g}$. Since $V(\mathbf{x})$ is $\mathcal{G}$-invariant, for all $g \in \mathcal{G}$ we have

$$V = V \circ L_g. \tag{33}$$

Now taking the gradient of both sides, and combining with the chain rule, we obtain

$$\nabla V = \nabla(V \circ L_g) = \mathbf{J}_{L_g}^\mathsf{T}(\nabla V \circ L_g). \tag{34}$$

Then by multiplying $\mathbf{J}_{L_g}$ on both sides, we have

$$\mathbf{J}_{L_g} \nabla V = \mathbf{J}_{L_g} \mathbf{J}_{L_g}^\mathsf{T}(\nabla V \circ L_g). \tag{35}$$

Considering $\mathcal{G}$ is a subgroup of the Euclidean group, the action of any group element $g \in \mathcal{G}$ on $\mathbb{R}^n$ can be represented by

$$L_g(\mathbf{x}) = A(\mathbf{x} + \mathbf{v}), \tag{36}$$

where $A \in O(n)$ is an orthogonal matrix. Therefore, $\mathbf{J}_{L_g} \mathbf{J}_{L_g}^\mathsf{T} = AA^\mathsf{T} = I$, where $I$ is an identity matrix, so we have $\mathbf{J}_{L_g} \nabla V = \nabla V \circ L_g$, which means that the curl-free vector field $\mathbf{f}_{curl} = \nabla V$ is $\mathcal{G}$-equivariant, as desired.

### A.4.2 Proof of Theorem 4.2

**Theorem 4.2.** *Let $\mathcal{G}$ be a Euclidean group or its subgroup, and let $\mathbf{v} : \mathbb{R}^n \to \mathbb{R}^n$ be a $\mathcal{G}$-equivariant vector field. Then the divergence-free vector field $\mathbf{f}_{div} = [\nabla \cdot \mathbf{A}_1, \ldots, \nabla \cdot \mathbf{A}_n]^\mathsf{T}$ is $\mathcal{G}$-equivariant, where $\mathbf{A}_i$ denotes the $i$-th row of the skew-symmetric matrix-valued function $\mathbf{A} = \mathbf{J}_\mathbf{v} - \mathbf{J}_\mathbf{v}^\mathsf{T}$.*

*Proof.* To prove this theorem, we need to show that the divergence-free vector field $\mathbf{f}_{div}$ satisfies the $\mathcal{G}$-equivariance condition, i.e., for all $g \in \mathcal{G}$ and $\mathbf{x} \in \mathbb{R}^n$,

$$\mathbf{f}_{div} \circ L_g = \mathbf{J}_{L_g} \mathbf{f}_{div}, \tag{37}$$

where $L_g(\mathbf{x}) : \mathbb{R}^n \to \mathbb{R}^n$ is the action of $g$ on $\mathbf{x}$, and $\mathbf{J}_{L_g}$ is the Jacobian matrix of $L_g$.

Define the divergence-free vector field $\mathbf{f}_{div} = [\nabla \cdot \mathbf{A}_1, ..., \nabla \cdot \mathbf{A}_n]^\mathsf{T}$, where $\mathbf{A}_i$ is the $i$th row of the skew-symmetric matrix field $\mathbf{A} = \mathbf{J}_\mathbf{v} - \mathbf{J}_\mathbf{v}^\mathsf{T}$, and $\mathbf{J}_\mathbf{v}$ is the Jacobian of some vector field $\mathbf{v} : \mathbb{R}^n \to \mathbb{R}^n$. Given that $\mathbf{v}$ is $\mathcal{G}$-equivariant, for all $g \in \mathcal{G}$ and $\mathbf{x} \in \mathbb{R}^n$ we have

$$\mathbf{v} \circ L_g = \mathbf{J}_{L_g} \mathbf{v}. \tag{38}$$

Now computing the Jacobian of both sides, by the chain rule of the Jacobian, we obtain

$$\mathbf{J}_{\mathbf{v} \circ L_g} = (\mathbf{J}_\mathbf{v} \circ L_g)\mathbf{J}_{L_g} = \mathbf{J}_{L_g}\mathbf{J}_\mathbf{v}. \tag{39}$$

Then applying $\mathbf{J}_{L_g}^\mathsf{T}$ on both sides, we have $\mathbf{J}_{L_g}^\mathsf{T}(\mathbf{J}_\mathbf{v} \circ L_g)\mathbf{J}_{L_g} = \mathbf{J}_{L_g}^\mathsf{T}\mathbf{J}_{L_g}\mathbf{J}_\mathbf{v}$. Since $\mathcal{G}$ is a subgroup of the Euclidean group, the Jacobian of the action of any group element $g \in \mathcal{G}$ on $\mathbb{R}^n$ is an orthogonal

matrix, i.e., $\mathbf{J}_{L_g} \in O(n)$. Therefore, $\mathbf{J}_{L_g}^\mathsf{T} \mathbf{J}_{L_g} = I$, where $I$ is an identity matrix, so it holds that $\mathbf{J}_\mathbf{v} = \mathbf{J}_{L_g}^\mathsf{T} (\mathbf{J}_\mathbf{v} \circ L_g) \mathbf{J}_{L_g}$. Then by substituting it into $\mathbf{A} = \mathbf{J}_\mathbf{v} - \mathbf{J}_\mathbf{v}^\mathsf{T}$, we obtain

$$\mathbf{A} = \mathbf{J}_\mathbf{v} - \mathbf{J}_\mathbf{v}^\mathsf{T} = \mathbf{J}_{L_g}^\mathsf{T} \left( \mathbf{J}_\mathbf{v} \circ L_g - \mathbf{J}_\mathbf{v}^\mathsf{T} \circ L_g \right) \mathbf{J}_{L_g} = \mathbf{J}_{L_g}^\mathsf{T} (\mathbf{A} \circ L_g) \mathbf{J}_{L_g}, \tag{40}$$

where the equation at the $(i, j)$-th entries of both sides is given by

$$\mathbf{A}_{ij} = \sum_{k=1}^{n} \left[ \frac{\partial (L_g)_k}{\partial x_i} \sum_{l=1}^{n} \frac{\partial (L_g)_l}{\partial x_j} (\mathbf{A}_{kl} \circ L_g) \right]. \tag{41}$$

Then by substituting $\mathbf{A}_{ij}$ into the construction of $\mathbf{f}_{div}$, for any $g \in \mathcal{G}$ and $\mathbf{x} \in \mathbb{R}^n$, we have

$$\mathbf{f}_{div} = [\nabla \cdot \mathbf{A}_1, \quad ..., \quad \nabla \cdot \mathbf{A}_n]^\mathsf{T} \tag{42}$$

$$= \left[ \sum_{j=1}^{n} \frac{\partial \mathbf{A}_{1j}}{\partial x_j}, \quad ..., \quad \sum_{j=1}^{n} \frac{\partial \mathbf{A}_{nj}}{\partial x_j} \right]^\mathsf{T} \tag{43}$$

$$= \begin{bmatrix} \sum_{j=1}^{n} \sum_{k=1}^{n} \frac{\partial (L_g)_k}{\partial x_1} \sum_{l=1}^{n} \frac{\partial (L_g)_l}{\partial x_j} \frac{\partial (\mathbf{A}_{kl} \circ L_g)}{\partial x_j} \\ \vdots \\ \sum_{j=1}^{n} \sum_{k=1}^{n} \frac{\partial (L_g)_k}{\partial x_n} \sum_{l=1}^{n} \frac{\partial (L_g)_l}{\partial x_j} \frac{\partial (\mathbf{A}_{kl} \circ L_g)}{\partial x_j} \end{bmatrix} \tag{44}$$

$$= \begin{bmatrix} \sum_{k=1}^{n} \frac{\partial (L_g)_k}{\partial x_1} \sum_{j=1}^{n} \sum_{l=1}^{n} \frac{\partial (L_g)_l}{\partial x_j} \frac{\partial (\mathbf{A}_{kl} \circ L_g)}{\partial x_j} \\ \vdots \\ \sum_{k=1}^{n} \frac{\partial (L_g)_k}{\partial x_n} \sum_{j=1}^{n} \sum_{l=1}^{n} \frac{\partial (L_g)_l}{\partial x_j} \frac{\partial (\mathbf{A}_{kl} \circ L_g)}{\partial x_j} \end{bmatrix} \tag{45}$$

$$= \begin{bmatrix} \sum_{k=1}^{n} \frac{\partial (L_g)_k}{\partial x_1} \sum_{j=1}^{n} \sum_{l=1}^{n} \frac{\partial (L_g)_l}{\partial x_j} \sum_{p=1}^{n} \frac{\partial (L_g)_p}{\partial x_j} \left( \frac{\partial \mathbf{A}_{kl}}{\partial x_p} \circ L_g \right) \\ \vdots \\ \sum_{k=1}^{n} \frac{\partial (L_g)_k}{\partial x_n} \sum_{j=1}^{n} \sum_{l=1}^{n} \frac{\partial (L_g)_l}{\partial x_j} \sum_{p=1}^{n} \frac{\partial (L_g)_p}{\partial x_j} \left( \frac{\partial \mathbf{A}_{kl}}{\partial x_p} \circ L_g \right) \end{bmatrix} \tag{46}$$

$$= \begin{bmatrix} \sum_{k=1}^{n} \frac{\partial (L_g)_k}{\partial x_1} \sum_{l=1}^{n} \sum_{p=1}^{n} \left( \frac{\partial \mathbf{A}_{kl}}{\partial x_p} \circ L_g \right) \sum_{j=1}^{n} \frac{\partial (L_g)_l}{\partial x_j} \frac{\partial (L_g)_p}{\partial x_j} \\ \vdots \\ \sum_{k=1}^{n} \frac{\partial (L_g)_k}{\partial x_n} \sum_{l=1}^{n} \sum_{p=1}^{n} \left( \frac{\partial \mathbf{A}_{kl}}{\partial x_p} \circ L_g \right) \sum_{j=1}^{n} \frac{\partial (L_g)_l}{\partial x_j} \frac{\partial (L_g)_p}{\partial x_j} \end{bmatrix} \tag{47}$$

$$= \left[ \sum_{k=1}^{n} \frac{\partial (L_g)_k}{\partial x_1} \sum_{l=1}^{n} \left( \frac{\partial \mathbf{A}_{kl}}{\partial x_l} \circ L_g \right), \quad ..., \quad \sum_{k=1}^{n} \frac{\partial (L_g)_k}{\partial x_n} \sum_{l=1}^{n} \left( \frac{\partial \mathbf{A}_{kl}}{\partial x_l} \circ L_g \right) \right]^\mathsf{T} \tag{48}$$

$$= \left[ \sum_{k=1}^{n} \frac{\partial (L_g)_k}{\partial x_1} (\nabla \cdot \mathbf{A}_k) \circ L_g, \quad ..., \quad \sum_{k=1}^{n} \frac{\partial (L_g)_k}{\partial x_n} (\nabla \cdot \mathbf{A}_k) \circ L_g \right]^\mathsf{T} \tag{49}$$

$$= \mathbf{J}_{L_g}^\mathsf{T} [(\nabla \cdot \mathbf{A}_1) \circ L_g, \quad ..., \quad (\nabla \cdot \mathbf{A}_n) \circ L_g]^\mathsf{T} \tag{50}$$

$$= \mathbf{J}_{L_g}^\mathsf{T} (\mathbf{f}_{div} \circ L_g). \tag{51}$$

Then, applying $\mathbf{J}_{L_g}$ on both sides, we obtain $\mathbf{J}_{L_g} \mathbf{f}_{div} = \mathbf{f}_{div} \circ L_g$, which completes the proof.

## A.5 Physical systems

We tested our method on three physical systems: a damped mass spring, a damped pendulum, and a Chua circuit. This section reviews these physical systems, listing their governing equations, HHDs and symmetry properties.

**Damped Mass Spring** A damped mass-spring system is a mass attached to a spring that oscillates periodically around an equilibrium position. Its Hamiltonian in natural units is given by

$$H(q, p) = \frac{1}{2} \left( q^2 + p^2 \right), \tag{52}$$

where $q \in \mathbb{R}$ is its position, and $p$ is the momentum conjugate to $q$. The Hamiltonian represents the total energy of the oscillator. Without energy dissipation, the motion of the oscillator is described

by the Hamiltonian vector field (div-free vector field $\mathbf{f}_{div}$) derived from the Hamiltonian:

$$\mathbf{f}_{div} := \begin{bmatrix} \dot{q} \\ \dot{p} \end{bmatrix} = \begin{bmatrix} \frac{\partial H}{\partial p} \\ -\frac{\partial H}{\partial q} \end{bmatrix} = \begin{bmatrix} p \\ -q \end{bmatrix}. \tag{53}$$

Then we show that the Hamiltonian vector field has SO(2)-equivariance. Apply a rotation by an angle $\theta$ that transforms any state vector $[q, p]^\mathsf{T}$ to $[q', p']^\mathsf{T}$, where $q' = q\cos\theta - p\sin\theta$, $p' = q\sin\theta + p\cos\theta$, then for all $\theta \in \mathbb{R}$, the equivariance condition (Eq. 16) can be easily verified by

$$\mathbf{f}_{div}(q', p') = \begin{bmatrix} p' \\ -q' \end{bmatrix} = \begin{bmatrix} \cos\theta & -\sin\theta \\ \sin\theta & \cos\theta \end{bmatrix} \begin{bmatrix} p \\ -q \end{bmatrix} = \begin{bmatrix} \cos\theta & -\sin\theta \\ \sin\theta & \cos\theta \end{bmatrix} \mathbf{f}_{div}(q, p). \tag{54}$$

Therefore, the Hamiltonian vector field is equivariant with respect to the 2D rotations. However, this rotation symmetry is broken when the system suffers from energy dissipation, $i.e.$, a dissipative field is added to the motion equations. The dissipative vector field (curl-free vector field $\mathbf{f}_{curl}$) is induced by the Rayleigh function $D(p) = \rho p^2/2$:

$$\mathbf{f}_{curl} = -\nabla_{(q,p)} D = [0, -\rho p]^\mathsf{T}, \tag{55}$$

where $\rho$ is the friction coefficient and we set it at 0.1. This dissipative field has the symmetries of the $q$-axis translation ($\mathbf{f}_{curl}(q + g, p) = \mathbf{f}_{curl}(q, p), \forall g \in \mathbb{R}$) and the rotation by an angle of $\pi$ radians (the so-called $odd\ symmetry$, $i.e.$, $\mathbf{f}_{curl}(-q, -p) = -\mathbf{f}_{curl}(q, p)$). Then by summing the curl-free vector field (Eq. 55) and the Hamiltonian vector field (Eq. 53), the dynamics of the damped harmonic oscillator is characterized by

$$\mathbf{f} = \mathbf{f}_{div} + \mathbf{f}_{curl} = [p, -q - \rho p]^\mathsf{T}, \tag{56}$$

which only demonstrates the odd symmetry.

**Damped Pendulum** A damped pendulum is a physical system consisting of a weight suspended from a pivot, subjected to a resistive force that gradually reduces its oscillation over time. By defining its state in the phase space through $[q, p]^\mathsf{T} \in \mathbb{R}^2$, the dynamics of a damped pendulum and its HHD can be given by

$$\mathbf{f} := \begin{bmatrix} \dot{q} \\ \dot{p} \end{bmatrix} = \begin{bmatrix} \frac{\partial H}{\partial p} \\ -\frac{\partial H}{\partial q} \end{bmatrix} - \begin{bmatrix} \frac{\partial D}{\partial q} \\ \frac{\partial D}{\partial p} \end{bmatrix} = \underbrace{\begin{bmatrix} \frac{l^2}{m}p \\ -2mgl\sin q \end{bmatrix}}_{\text{divergence} - \text{free}} + \underbrace{\begin{bmatrix} 0 \\ -\rho p \end{bmatrix}}_{\text{curl} - \text{free}} = \begin{bmatrix} \frac{l^2}{m}p \\ -2mgl\sin q - \rho p \end{bmatrix}, \tag{57}$$

where the Hamiltonian $H$ and the Rayleigh function $D$ are given by

$$H(q, p) = 2mgl(1 - \cos q) + \frac{l^2 p^2}{2m}, \ D(p) = \frac{1}{2}\rho p^2. \tag{58}$$

In our experiments, the gravitational constant $g$, the mass $m$, and the pendulum length $l$ were set as $g = 3$ and $m = l = 1$, and the friction coefficient $\rho$ was set as $\rho = 0.1$. The dynamics of the damped pendulum, as well as its div-free and curl-free components, exhibit the odd symmetry, but the curl-free part additionally exhibits the translation invariance along the $q$-axis.

**Chua Circuit** The Chua circuit (Matsumoto, 1984) is a simple electronic circuit that exhibits chaotic behavior, and it has applications in various fields, such as secure communication systems and random number generators. The ODE of a Chua circuit with its HHD is given by

$$\begin{bmatrix} \dot{x} \\ \dot{y} \\ \dot{z} \end{bmatrix} = \begin{bmatrix} \alpha(y - x^3 - cx) \\ x - y + z \\ -\beta y \end{bmatrix} = \underbrace{\begin{bmatrix} \alpha y \\ x + z \\ -\beta y \end{bmatrix}}_{\text{divergence} - \text{free}} + \underbrace{\begin{bmatrix} \alpha(-x^3 - cx) \\ -y \\ 0 \end{bmatrix}}_{\text{curl} - \text{free}}, \tag{59}$$

where $x, y, z$ are the phase space variables, and $\alpha, c, \beta$ are the system parameters, which were set to 10, $-0.143$, and 16 respectively in the experiments. The Chua circuit is equivariant under a $\pi$ rotation about the origin: $L_\pi : (x, y, z) \mapsto (-x, -y, -z)$. If the Chua circuit is decomposed by the HHD in Eq. 59, more symmetries are exhibited. Specifically, its div-free component inherits the odd

symmetry, but additionally presents the translation invariance along $x = z$. The curl-free vector field is invariant along the $z$-axis and equivariant under mirror reflections across the coordinate planes,

$$L_{i,j} : (x, y, z) \mapsto \left( (-1)^i x, (-1)^j y, z \right), \forall i, j \in \{0, 1\}. \tag{60}$$

This HHD not only uncovers more knowledge of the Chua circuit's symmetry but can also be used to analysis the energy of the system. According to the generalized Hamiltonian formalism (Eq. 22), the energy function $H$ associated with the Chua circuit satisfies the following PDE:

$$\alpha y \frac{\partial H}{\partial x} + (x + z) \frac{\partial H}{\partial y} - \beta y \frac{\partial H}{\partial z} = 0, \tag{61}$$

which is satisfied by the quadratic form:

$$H = \frac{1}{2} \left( -\frac{1}{\alpha} x^2 + y^2 + \frac{1}{\beta} z^2 \right). \tag{62}$$

As presented by Zhou et al. (2021), this energy function indicates that the Chua circuit keeps oscillatory when the energy release along the $x$-axis is enough to balance the energy pumping along the $y$-axis and $z$-axis.

### A.6 Uniqueness and Symmetries

The two constituent kernels of our model define the space of divergence-free vector fields ($\mathbf{f}_{div} \in \mathcal{F}_{div}$) and the space of curl-free vector fields ($\mathbf{f}_{curl} \in \mathcal{F}_{curl}$), respectively. These two spaces overlap partially due to the presence of harmonic vector fields ($\mathbf{f}_{harm} \in \mathcal{F}_{div} \cap \mathcal{F}_{curl}$). To eliminate this overlap and make HHD unique, we propose to impose symmetry constraints on the two spaces separately, with the corresponding symmetry groups defined as $\mathcal{G}_{div}$ and $\mathcal{G}_{curl}$. Therefore, the uniqueness property of HHD depends on the space of harmonic vector fields that respects the union of two symmetry groups, $i.e.$

$$\mathcal{F}_{harm} = \left\{ \mathbf{f}_{harm} \mid \mathbf{f}_{harm} \circ L_g = \mathbf{J}_{L_g} \mathbf{f}_{harm}, \forall \mathbf{x} \in \mathbb{R}^n, g \in \mathcal{G}_{div} \cup \mathcal{G}_{curl} \right\}. \tag{63}$$

If $\mathcal{F}_{harm} = \emptyset$, the HHD is unique, $i.e.$ our model is identifiable. A harmonic vector field can always be represented by the gradient field of a harmonic function $h$, which is a scalar function satisfying $\nabla \cdot \nabla h = 0$ (Laplace's equation). For the three dynamical systems used in our experiments (detailed in Appendix A.5), it can be easily verified that the presence of harmonic components can be eliminated through forced symmetries. The symmetry group union of a damped mass-spring system consists of a rotation group $\mathcal{G}_{div} = SO(2)$ and a translation group $\mathcal{G}_{curl} = \{(g, 0) \mid g \in \mathbb{R}\}$. A harmonic vector field $\mathbf{f}_{harm} = \nabla h$ that satisfies this translation symmetry implies that its harmonic function $h(q, p)$ is independent of the variable $q$, so the harmonic vector field is given by $\mathbf{f}_{harm} = (0, \partial_p h)$ and $\partial_p h$ should be a constant to satisfy the Laplace's equation, but the harmonic vector field in the form of constant clearly contradicts rotation symmetry $\mathcal{G}_{div}$. Therefore, there is no harmonic vector field that respects both the symmetry groups $\mathcal{G}_{div}$ and $\mathcal{G}_{curl}$. Similar conclusions can be drawn for the damped pendulum and the Chua circuit. The Laplace's equation and translation symmetry imply that harmonic vector fields can only exist in the form of constant vector fields. However, constant vector fields obviously contradict odd symmetry or mirror symmetry.

### A.7 Implementation details

The experiments were performed on a single Nvidia GeForce GTX 3050 Ti GPU, and all of the models were implemented with PyTorch (Paszke et al., 2019). The GP-based models (Ind-GP, GIM-GP, Div-GP, HHD-GP and SPHHD-GP) were trained by maximizing the log of their marginal likelihood (Eq. 3):

$$\log p(\mathbf{Y} \mid \mathbf{X}) = \log \mathcal{N}(\mathbf{Y} \mid \mathbf{0}, \mathbf{K} + \Sigma) = -\frac{1}{2} \mathbf{Y}^\top (\mathbf{K} + \Sigma)^{-1} \mathbf{Y} - \frac{1}{2} \log |\mathbf{K} + \Sigma| - m \log 2\pi, \tag{64}$$

where $|\cdot|$ computes the determinant of the covariance matrix $\mathbf{K} + \Sigma$. Training a GP model refers to optimizing the kernel parameters in $\mathbf{K}$ and the noise variances in $\Sigma$, and these hyperparameters were initialized randomly in our experiments. The GP-based models were trained by the ADAM optimizer (Kingma & Ba, 2014), with a learning rate $0.01$ for $3000$ gradient steps. The kernels of the GP-based models are constructed as follows.

### A.7.1    IND-GP

The Ind-GP models each dimension of a $n$-dimensional dynamical system independently, so its matrix-valued kernel is constructed by

$$\kappa_{ind} = \text{diag}\left(\kappa_1, \ldots, \kappa_n\right), \tag{65}$$

where $\kappa_i$, $i = 1, \ldots, n$ are independent scalar kernels. And a standard choice for $\kappa_i$ is the squared exponential (SE) kernel,

$$\kappa_{se}\left(\mathbf{x}, \mathbf{x}'\right) = \sigma^2 \exp\left(-\frac{1}{2}l^{-2}\left\|\mathbf{x} - \mathbf{x}'\right\|^2\right), \tag{66}$$

which has two parameters: $\sigma$ determines the variation of function values from their mean, and $l$ controls the length scale on which the function varies. Realizations of GPs with SE kernels are dense in the set of smooth functions $C^\infty\left(\mathbb{R}^n, \mathbb{R}\right)$ (cf. Prop.1 in Lange-Hegermann & Robertz (2022)). For a fair comparison, we also used the SE kernel (Eq. 66) to construct kernels for the other GP-based models.

### A.7.2    GIM-GP

The GIM-GP produces predictions with the desired symmetry. Each of the systems in our experiments exhibits the odd symmetry as a whole. So their symmetry groups can be given by $\mathcal{G}_{odd} = \{I_n, -I_n\}$, where $I_n$ is the $n$-dimensional identity matrix, and the group elements are linear representations of the group actions, $i.e.$, $L_g\left(\mathbf{x}\right) = g\mathbf{x}$, $\forall g \in \mathcal{G}$. Therefore, according to Eq. 19, the GIM-kernel for $\mathcal{G}_{odd}$ was constructed by

$$\kappa_{gim} = \left(\kappa_{se}\left(\mathbf{x}, \mathbf{x}'\right) - \kappa_{se}\left(\mathbf{x}, -\mathbf{x}'\right)\right)I_n, \tag{67}$$

where the SE kernel (Eq. 66) was used as the basis kernel $\kappa$ in Eq. 19.

### A.7.3    DIV-GP

GPs with a div-free kernel can be used to approximate conservative dynamics. According to Eq. 14, a two-dimensional div-free kernel was constructed by

$$\kappa_{div} = \begin{bmatrix} \frac{\partial^2 \kappa_H\left(\mathbf{x}, \mathbf{x}'\right)}{\partial x_2 \partial x_2'} & -\frac{\partial^2 \kappa_H\left(\mathbf{x}, \mathbf{x}'\right)}{\partial x_2 \partial x_1'} \\ -\frac{\partial^2 \kappa_H\left(\mathbf{x}, \mathbf{x}'\right)}{\partial x_1 \partial x_2'} & \frac{\partial^2 \kappa_H\left(\mathbf{x}, \mathbf{x}'\right)}{\partial x_1 \partial x_1'} \end{bmatrix}, \tag{68}$$

where $\kappa_{\mathbf{u}}$ in Eq. 14 is denoted by $\kappa_H$ instead, indicating that it is the kernel for the Hamiltonian functions of the damped mass spring and the damped pendulum, and the SE kernel (Eq. 66) was used for $\kappa_H$. The partial derivatives in Eq. 68 were calculated by automatic differentiation in PyTorch, so we did not need to derive its analytic expression.

Then we constructed the div-free kernel for the Chua circuit system. According to Eq. 13, a three-dimensional div-free vector field is given by $\mathbf{f}_{div}\left(\mathbf{x}\right) = \Psi\mathbf{u}\left(\mathbf{x}\right)$, where $\Psi = [\psi_{12}, \psi_{13}, \psi_{23}] \in R^{3\times3}$ with its components given by

$$\psi_{12} = \begin{bmatrix} 0 & 1 & 0 \\ -1 & 0 & 0 \\ 0 & 0 & 0 \end{bmatrix}\begin{bmatrix} \partial_{x_1} \\ \partial_{x_2} \\ \partial_{x_3} \end{bmatrix} = \begin{bmatrix} \partial_{x_2} \\ -\partial_{x_1} \\ 0 \end{bmatrix}, \tag{69}$$

$$\psi_{13} = \begin{bmatrix} 0 & 0 & 1 \\ 0 & 0 & 0 \\ -1 & 0 & 0 \end{bmatrix}\begin{bmatrix} \partial_{x_1} \\ \partial_{x_2} \\ \partial_{x_3} \end{bmatrix} = \begin{bmatrix} \partial_{x_3} \\ 0 \\ -\partial_{x_1} \end{bmatrix}, \tag{70}$$

$$\psi_{23} = \begin{bmatrix} 0 & 0 & 0 \\ 0 & 0 & 1 \\ 0 & -1 & 0 \end{bmatrix}\begin{bmatrix} \partial_{x_1} \\ \partial_{x_2} \\ \partial_{x_3} \end{bmatrix} = \begin{bmatrix} 0 \\ \partial_{x_3} \\ -\partial_{x_2} \end{bmatrix}. \tag{71}$$

Then by assuming $\mathbf{u} = [u_{12}, u_{13}, u_{23}]^\mathsf{T} \sim \mathcal{GP}\left(\mathbf{0}, \kappa_\mathbf{u} = \kappa_{ind}\right)$, we constructed the div-free kernel for the Chua circuit according to Eq. 14:

$$\kappa_{div} = \begin{bmatrix} \partial_{x_2} & \partial_{x_3} & 0 \\ -\partial_{x_1} & 0 & \partial_{x_3} \\ 0 & -\partial_{x_1} & -\partial_{x_2} \end{bmatrix} \begin{bmatrix} \kappa_1 & 0 & 0 \\ 0 & \kappa_2 & 0 \\ 0 & 0 & \kappa_3 \end{bmatrix} \begin{bmatrix} \partial_{x_2} & -\partial_{x_1} & 0 \\ \partial_{x_3} & 0 & -\partial_{x_1} \\ 0 & \partial_{x_3} & -\partial_{x_2} \end{bmatrix} \tag{72}$$

$$= \begin{bmatrix} \frac{\partial^2 \kappa_1(\mathbf{x},\mathbf{x}')}{\partial x_2 \partial x_2'} + \frac{\partial^2 \kappa_2(\mathbf{x},\mathbf{x}')}{\partial x_3 \partial x_3'} & -\frac{\partial^2 \kappa_1(\mathbf{x},\mathbf{x}')}{\partial x_2 \partial x_1'} & -\frac{\partial^2 \kappa_2(\mathbf{x},\mathbf{x}')}{\partial x_3 \partial x_1'} \\ -\frac{\partial^2 \kappa_1(\mathbf{x},\mathbf{x}')}{\partial x_1 \partial x_2'} & \frac{\partial^2 \kappa_1(\mathbf{x},\mathbf{x}')}{\partial x_1 \partial x_1'} + \frac{\partial^2 \kappa_3(\mathbf{x},\mathbf{x}')}{\partial x_3 \partial x_3'} & -\frac{\partial^2 \kappa_3(\mathbf{x},\mathbf{x}')}{\partial x_3 \partial x_2'} \\ -\frac{\partial^2 \kappa_2(\mathbf{x},\mathbf{x}')}{\partial x_1 \partial x_3'} & -\frac{\partial^2 \kappa_3(\mathbf{x},\mathbf{x}')}{\partial x_2 \partial x_3'} & \frac{\partial^2 \kappa_2(\mathbf{x},\mathbf{x}')}{\partial x_1 \partial x_1'} + \frac{\partial^2 \kappa_3(\mathbf{x},\mathbf{x}')}{\partial x_2 \partial x_2'} \end{bmatrix}, \tag{73}$$

where $\kappa_1$, $\kappa_2$ and $\kappa_3$ are all independent SE kernels (Eq. 66).

### A.7.4 HHD-GP

The HHD-GP consists of two independent GPs added together, modeling curl-free and div-free dynamics respectively. GPs with div-free kernels have been constructed earlier, so here we build curl-free kernels according to Eq. 9. For the damped mass spring and the damped pendulum, their two-dimensional curl-free kernel was constructed by

$$\kappa_{curl} = \begin{bmatrix} \frac{\partial^2 \kappa_V(\mathbf{x},\mathbf{x}')}{\partial x_1 \partial x_1'} & \frac{\partial^2 \kappa_V(\mathbf{x},\mathbf{x}')}{\partial x_1 \partial x_2'} \\ \frac{\partial^2 \kappa_V(\mathbf{x},\mathbf{x}')}{\partial x_2 \partial x_1'} & \frac{\partial^2 \kappa_V(\mathbf{x},\mathbf{x}')}{\partial x_2 \partial x_2'} \end{bmatrix}. \tag{74}$$

Similarly, the three-dimensional curl-free kernel for the Chua circuit was constructed by

$$\kappa_{curl} = \begin{bmatrix} \frac{\partial^2 \kappa_V(\mathbf{x},\mathbf{x}')}{\partial x_1 \partial x_1'} & \frac{\partial^2 \kappa_V(\mathbf{x},\mathbf{x}')}{\partial x_1 \partial x_2'} & \frac{\partial^2 \kappa_V(\mathbf{x},\mathbf{x}')}{\partial x_1 \partial x_3'} \\ \frac{\partial^2 \kappa_V(\mathbf{x},\mathbf{x}')}{\partial x_2 \partial x_1'} & \frac{\partial^2 \kappa_V(\mathbf{x},\mathbf{x}')}{\partial x_2 \partial x_2'} & \frac{\partial^2 \kappa_V(\mathbf{x},\mathbf{x}')}{\partial x_2 \partial x_3'} \\ \frac{\partial^2 \kappa_V(\mathbf{x},\mathbf{x}')}{\partial x_3 \partial x_1'} & \frac{\partial^2 \kappa_V(\mathbf{x},\mathbf{x}')}{\partial x_3 \partial x_2'} & \frac{\partial^2 \kappa_V(\mathbf{x},\mathbf{x}')}{\partial x_3 \partial x_3'} \end{bmatrix}, \tag{75}$$

where $\kappa_V$ is a SE kernel.

### A.7.5 SPHHD-GP

**Symmetry-preserving curl-free kernels** According to Theorem 4.1, a $\mathcal{G}$-equivariant curl-free vector field is constructed by constraining its scalar potential $V$ to be $\mathcal{G}$-invariant. Therefore, a $\mathcal{G}$-equivariant curl-free kernel is obtained by constructing its potential kernel $\kappa_V$ according to Eq. 18.

The curl-free vector field (Eq. 55) of the damped mass spring and the damped pendulum has two types of symmetry:

*(i)* translation along the $q$-axis, *i.e.*, $L_g(\mathbf{x}) = \mathbf{x} + (g, 0)$, for all $g \in \mathbb{R}$;

*(ii)* rotation by an angle of $\pi$ radians, *i.e.*, $L_g(\mathbf{x}) = g\mathbf{x}$, for all $g \in \{I_2, -I_2\}$.

These symmetry groups were enforced to a SE kernel sequentially according to Eq. 18, where $|\mathcal{G}|$ can be ignored. Specifically, the translation invariance was enforced using the Gaussian integral formula:

$$\kappa_V = \int_{-\infty}^{+\infty} \kappa_{se}\left(\mathbf{x}, \mathbf{x}' + [g, 0]^\mathsf{T}\right) dg \tag{76}$$

$$= \kappa_{se}\left(p, p'\right) \int_{-\infty}^{+\infty} \exp(-\frac{(q - q' - g)^2}{2l^2}) dg \tag{77}$$

$$= \sqrt{2\pi} l \kappa_{se}\left(p, p'\right), \tag{78}$$

then based on which the $\pi$ rotation invariance was enforced by

$$\kappa_V = \sqrt{2\pi} l \left(\kappa_{se}\left(x_2, x_2'\right) + \kappa_{se}\left(x_2, -x_2'\right)\right). \tag{79}$$

The curl-free vector field of the Chua circuit (Eq. 59) has similar two types of symmetry:

*(i)* translation along the $z$-axis, *i.e.*, $L_g(\mathbf{x}) = \mathbf{x} + (0, 0, g)$, for all $g \in \mathbb{R}$;

*(ii)* mirror reflections across the coordinate planes, *i.e.*, $L_{i,j} : (x, y, z) \mapsto \left( (-1)^i x, (-1)^j y, z \right)$, for all $i, j \in \{0, 1\}$.

Therefore, $\kappa_V$ for the Chua circuit (Eq. 59) was constructed by

$$\kappa_V = \sqrt{2\pi} l \sum_{g \in \mathcal{G}} \kappa_{se}(\mathbf{x}, g\mathbf{x}'), \ \mathcal{G} = \left\{ \mathrm{diag}\left( (-1)^i, (-1)^j, 0 \right) \mid i, j \in \{0, 1\} \right\}. \tag{80}$$

**Symmetry-preserving div-free kernels**  Theorem 4.2 shows that a $\mathcal{G}$-equivariant div-free vector field is constructed from a vector potential $\mathbf{v}$ with the same equivariance. For the div-free vector field of the damped mass-spring system with SO(2)-equivariance, its $\kappa_\mathbf{v}$ was constructed by the GIM-kernel (Eq. 19):

$$\kappa_\mathbf{v}(\mathbf{x}, \mathbf{x}') = \int_0^{2\pi} \kappa_{se}(\mathbf{x}, g_\theta \mathbf{x}') g_\theta d\theta, \ \text{where } g_\theta = \begin{bmatrix} \cos\theta & -\sin\theta \\ \sin\theta & \cos\theta \end{bmatrix}. \tag{81}$$

Then this $\kappa_\mathbf{v}$ (Eq. 81) is substituted into Eq. 20 to construct $\kappa_H$ for its div-free kernel $\kappa_{div}$ (Eq. 68):

$$\kappa_H = \frac{\partial^2}{\partial x_2 \partial x_2'} [\kappa_\mathbf{v}]_{1,1} + \frac{\partial^2}{\partial x_1 \partial x_1'} [\kappa_\mathbf{v}]_{2,2} - \frac{\partial^2}{\partial x_2 \partial x_1'} [\kappa_\mathbf{v}]_{1,2} - \frac{\partial^2}{\partial x_1 \partial x_2'} [\kappa_\mathbf{v}]_{2,1} \tag{82}$$

$$= \int_0^{2\pi} \kappa_{se}(\mathbf{x}, g_\theta \mathbf{x}') \frac{2l^2 - \|\mathbf{x} - g_\theta \mathbf{x}'\|^2}{l^4} d\theta. \tag{83}$$

which admits no closed-form solution, so we used a numerical approximation of the integral (Eq. 83) by sampling discrete rotations of $\left\{ \theta = \frac{\pi}{4} n \mid n = 0, \ldots, 7 \right\}$, whose rotation matrices form a finite group. And by setting $\theta \in \{0, \pi\}$, Eq. 83 was used to construct $\kappa_H$ for the div-free vector field of the damped pendulum, which has the equivariance under a $\pi$ rotation (odd symmetry).

The div-free vector field of the Chui circuit (Eq. 59) has the odd symmetry and the translation symmetry along $x = z$. $\kappa_\mathbf{v}$ respecting the translation symmetry was constructed by $\kappa_\mathbf{v} = \kappa \cdot I_3$, where $\kappa$ is given by

$$\kappa = \int_{-\infty}^{+\infty} \kappa_{se}\left( [x, y, z]^\mathsf{T}, [x', y', z']^\mathsf{T} + [g, 0, -g]^\mathsf{T} \right) dg \tag{84}$$

$$= \kappa_{se}(y, y') \int_{-\infty}^{+\infty} \exp\left( -\frac{(x - x' - g)^2 + (z - z' + g)^2}{2l^2} \right) dg \tag{85}$$

$$= \kappa_{se}(y, y') \int_{-\infty}^{+\infty} \exp\left( -\frac{2\left( g - \frac{x_1 - x_1' - x_3 + x_3'}{2} \right)^2 + \frac{(x_1 - x_1' + x_3 - x_3')^2}{2}}{2l^2} \right) dg \tag{86}$$

$$= \sqrt{\pi} l \kappa_{se}(x_2, x_2') \int_{-\infty}^{+\infty} \exp\left( -\frac{(x_1 - x_1' + x_3 - x_3')^2}{4l^2} \right) dg \tag{87}$$

$$= \sqrt{\pi} l \kappa_{se}(\mathbf{p}, \mathbf{p}'). \tag{88}$$

where $\mathbf{p} = \left[ \frac{1}{\sqrt{2}}(x + z), y \right]^\mathsf{T}$. And we further enforced the odd symmetry by

$$\kappa = \sqrt{\pi} l \left( \kappa_{se}(\mathbf{p}, \mathbf{p}') - \kappa_{se}(\mathbf{p}, -\mathbf{p}') \right). \tag{89}$$

Then, $\kappa_\mathbf{v} = \kappa \cdot I_3$ was substituted into Eq. 20 to construct $\kappa_\mathbf{u}$ by

$$\kappa_\mathbf{u} = \begin{bmatrix} \frac{\partial^2 \kappa(\mathbf{x}, \mathbf{x}')}{\partial x_2 \partial x_2'} + \frac{\partial^2 \kappa(\mathbf{x}, \mathbf{x}')}{\partial x_1 \partial x_1'} & \frac{\partial^2 \kappa(\mathbf{x}, \mathbf{x}')}{\partial x_2 \partial x_3'} & -\frac{\partial^2 \kappa(\mathbf{x}, \mathbf{x}')}{\partial x_1 \partial x_3'} \\ \frac{\partial^2 \kappa(\mathbf{x}, \mathbf{x}')}{\partial x_3 \partial x_2'} & \frac{\partial^2 \kappa(\mathbf{x}, \mathbf{x}')}{\partial x_3 \partial x_3'} + \frac{\partial^2 \kappa(\mathbf{x}, \mathbf{x}')}{\partial x_1 \partial x_1'} & \frac{\partial^2 \kappa(\mathbf{x}, \mathbf{x}')}{\partial x_1 \partial x_2'} \\ -\frac{\partial^2 \kappa(\mathbf{x}, \mathbf{x}')}{\partial x_3 \partial x_1'} & \frac{\partial^2 \kappa(\mathbf{x}, \mathbf{x}')}{\partial x_2 \partial x_1'} & \frac{\partial^2 \kappa(\mathbf{x}, \mathbf{x}')}{\partial x_3 \partial x_3'} + \frac{\partial^2 \kappa(\mathbf{x}, \mathbf{x}')}{\partial x_2 \partial x_2'} \end{bmatrix}. \tag{90}$$

Finally, this $\kappa_\mathbf{u}$ (Eq. 90) was substituted into Eq. 14 to construct the symmetry-preserving div-free kernel for the Chua circuit.

### A.7.6  D-HNN

We used the released code[5] of D-HNN and ran their training routine for our systems.

### A.7.7  Implementation Details of Predicting Energy

**The damped mass-spring system and the damped pendulum**  When HHD-GP and SPHHD-GP are used to learn the damped mass-spring system and the damped pendulum, the potential kernel $\kappa_H$ for constructing their div-free kernels (Eq. 68) can be interpreted as placing a GP prior on the Hamiltonian function. Therefore, a joint GP describing both the Hamiltonian function $H(\mathbf{x})$ and the dynamical system $\mathbf{f}(\mathbf{x})$ is given by

$$\begin{bmatrix} H(\mathbf{x}) \\ \mathbf{f}(\mathbf{x}) \end{bmatrix} \sim \mathcal{GP} \left( \begin{bmatrix} 0 \\ \mathbf{0} \end{bmatrix}, \begin{bmatrix} \kappa_H(\mathbf{x},\mathbf{x}') & \kappa_{H,\mathbf{f}}(\mathbf{x},\mathbf{x}') \\ \kappa_{\mathbf{f},H}(\mathbf{x},\mathbf{x}') & \kappa_{hhd}(\mathbf{x},\mathbf{x}') \end{bmatrix} \right), \tag{91}$$

where $\kappa_{H,\mathbf{f}}(\mathbf{x},\mathbf{x}') = \kappa_{\mathbf{f},H}(\mathbf{x}',\mathbf{x})^\mathsf{T}$ with

$$\kappa_{H,\mathbf{f}}(\mathbf{x},\mathbf{x}') = \text{cov}\left[H(\mathbf{x}),\mathbf{f}(\mathbf{x}')\right] = [\partial_{p'},-\partial_{q'}]\,\kappa_H(\mathbf{x},\mathbf{x}'); \tag{92}$$

$$\kappa_{\mathbf{f},H}(\mathbf{x},\mathbf{x}') = \text{cov}\left[\mathbf{f}(\mathbf{x}),H(\mathbf{x}')\right] = [\partial_p,-\partial_q]^\mathsf{T}\,\kappa_H(\mathbf{x},\mathbf{x}'). \tag{93}$$

After training the model $\mathbf{f}(\mathbf{x}) \sim \mathcal{GP}(\mathbf{0},\kappa_{hhd}(\mathbf{x},\mathbf{x}'))$ on noisy observations $\mathbf{Y} = [\mathbf{y}_1,\ldots,\mathbf{y}_m]^{\mathsf{T}[6]}$ at states $\mathbf{X} = [\mathbf{x}_1,\ldots,\mathbf{x}_m]^\mathsf{T}$, we are interested in predicting the value of Hamiltonian function $H(\mathbf{x}_*)$ at a new test state $\mathbf{x}_*$. Since these data determine $H(\cdot)$ only up to an additive constant, we assume that we have an anchor point $H(\mathbf{x}_0)$ which can be chosen arbitrarily. Then, according to the GP prior (Eq. 91), $H(\mathbf{x}_*)$ and $\mathbf{Y}_H = [H(\mathbf{x}_0),\mathbf{Y}]^\mathsf{T}$ are jointly distributed as

$$\begin{bmatrix} H(\mathbf{x}_*) \\ \mathbf{Y}_H \end{bmatrix} \sim \mathcal{N}\left( \begin{bmatrix} 0 \\ \mathbf{0} \end{bmatrix}, \begin{bmatrix} \kappa_H(\mathbf{x}_*,\mathbf{x}_*) & \mathbf{k} \\ \mathbf{k}^\mathsf{T} & \mathbf{K} \end{bmatrix} \right), \tag{94}$$

where $\mathbf{k} = [\kappa_H(\mathbf{x}_*,\mathbf{x}_0),\kappa_{H,\mathbf{f}}(\mathbf{x}_*,\mathbf{X})]$, and

$$\mathbf{K} = \begin{bmatrix} \kappa_H(\mathbf{x}_0,\mathbf{x}_0) & \kappa_{H,\mathbf{f}}(\mathbf{x}_0,\mathbf{X}) \\ \kappa_{\mathbf{f},H}(\mathbf{X},\mathbf{x}_0) & \kappa_{hhd}(\mathbf{X},\mathbf{X}) + \sigma I \end{bmatrix}. \tag{95}$$

Then, we obtain the posterior distribution

$$p\left(H(\mathbf{x}_*)\mid\mathbf{Y}_H\right) = \mathcal{N}\left(\mathbf{k}\mathbf{K}^{-1}\mathbf{Y}_H,\ \kappa_H(\mathbf{x}_*,\mathbf{x}_*) - \mathbf{k}\mathbf{K}^{-1}\mathbf{k}^\mathsf{T}\right), \tag{96}$$

where the mean function is used for energy prediction.

**The Chua circuit**  For the Chua circuit, our model has no direct access to its Hamiltonian function, but we can estimate it using the generalized Hamiltonian formalism (Eq. 22), which states that the divergence-free vector field is always orthogonal to the gradient of the Hamiltonian function, *i.e.*, $\nabla H^\mathsf{T} \mathbf{f}_{div}(\mathbf{x}) = 0$, for all $\mathbf{x} \in \mathbb{R}^n$. We parameterize the Hamiltonian of the Chua circuit in a quadratic form: $\hat{H}(\mathbf{x}) = \frac{1}{2}\left(a_1 x^2 + a_2 y^2 + a_3 z^2\right)$, where $\mathbf{a} = [a_1, a_2, a_3]$ are parameters. Then by learning a divergence-free vector field $\hat{\mathbf{f}}_{div}(\cdot)$ through our model, we can estimate the parameters $\mathbf{a}$ by minimizing $\sum_{i=1}^m \left(\nabla\hat{H}^\mathsf{T}\hat{\mathbf{f}}(\mathbf{x}_i)\right)^2$ at a finite number of sample points $\{\mathbf{x}_i\}_{i=1}^m$ ($m = 500$ in our experiments). And to eliminate the solution at $\mathbf{a} = [0,0,0]$, we add an equality constraint at a random point of the ground truth $\{\mathbf{x}_0, H(\mathbf{x}_0)\}$ (Eq. 62). Therefore, the parameters $\mathbf{a}$ of $\hat{H}$ are solved in a convex quadratic program (QP):

$$\begin{aligned} \min\quad & \mathbf{a}^T\mathbf{Q}\mathbf{a} \\ \text{s.t.}\quad & \hat{H}(\mathbf{x}_0) = H(\mathbf{x}_0) \end{aligned} \tag{97}$$

where $\mathbf{Q} = \mathbf{q}^\mathsf{T}\mathbf{q}$ with $\mathbf{q} = \left[\hat{\mathbf{f}}_{div}(\mathbf{x}_1),\ldots,\hat{\mathbf{f}}_{div}(\mathbf{x}_m)\right]^\mathsf{T} \in \mathbb{R}^{m\times 3}$. And in the experiments, we solved the QP using the solver provided by CVXOPT (Andersen et al., 2020).

---

[5] https://github.com/DrewSosa/dissipative_hnns

[6] $\mathbf{y}_i = \mathbf{f}(\mathbf{x}_i) + \epsilon, \epsilon \overset{\text{i.i.d}}{\sim} \mathcal{N}(\mathbf{0},\sigma I)$

## A.8 EXPERIMENT RESULTS

### A.8.1 RESULTS OF ENERGY PREDICTION

To compare the accuracy of predicting the energy evolution along a state trajectory $\{\mathbf{x}_1, \ldots, \mathbf{x}_p\}$, we compute its RMSE by

$$\mathrm{RMSE}_{\mathrm{en}}\left(\hat{H}, H\right) = \left(\frac{1}{p}\sum_{i=0}^{p}\left(\hat{H}\left(\mathbf{x}_i\right) - H\left(\mathbf{x}_i\right)\right)^2\right)^{\frac{1}{2}}, \tag{98}$$

where $\hat{H}\left(\cdot\right)$ and $H\left(\cdot\right)$ are the predicted and true Hamiltonian function, respectively. And we averaged this RMSE over trajectories integrated from 50 random initial conditions, and the implementation details for generating these trajectories are the same as those for calculating the VPT (detailed in Section 6). The results of RMSE of energy prediction are shown in Table 2.

Table 2: Performance comparison of different methods on the energy prediction tasks for each dynamical system. The RMSE is recorded in the scale of $\times 10^{-2}$ and in the form of mean $\pm$ standard deviation. Bold font indicates best results.

| Physical System | D-HNN | HHD-GP | SPHHD-GP |
|---|---|---|---|
| Damped Mass Spring | $16.76 \pm 10.73$ | $33.34 \pm 15.78$ | $\mathbf{0.15 \pm 0.12}$ |
| Damped pendulum | $94.59 \pm 24.22$ | $43.18 \pm 24.68$ | $\mathbf{0.79 \pm 0.29}$ |
| Chua Circuit | N/A | $9.52 \pm 5.03$ | $\mathbf{0.13 \pm 0.14}$ |

### A.8.2 RESULTS OF TRAJECTORY PREDICTION

Fig. 5, Fig. 6, and Fig. 7 show the trajectory predictions for the damped mass-spring system, the damped pendulum, and the Chua circuit, respectively.

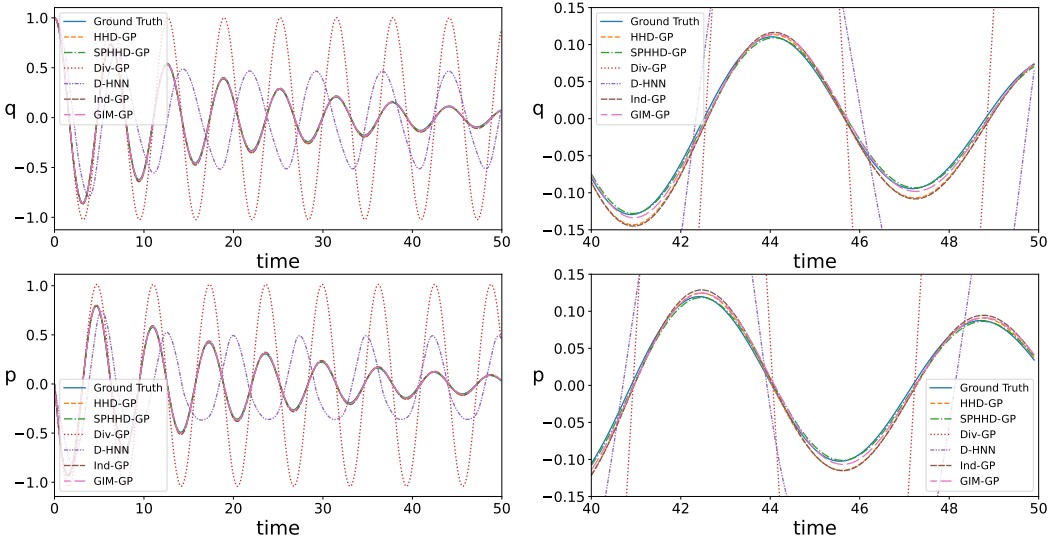

Figure 5: Comparison of predicted trajectories of the mass-spring system.

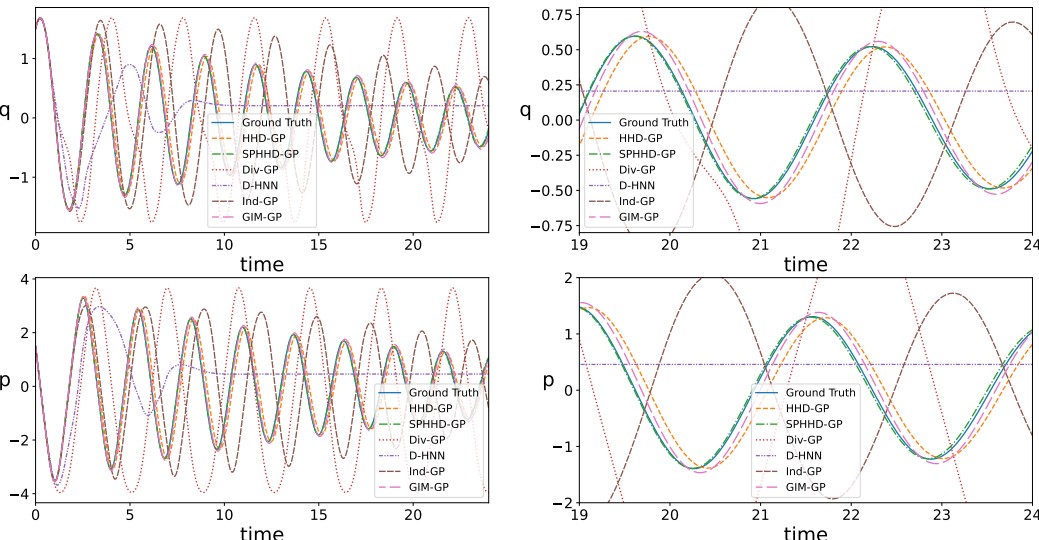

Figure 6: Comparison of predicted trajectories of the pendulum.

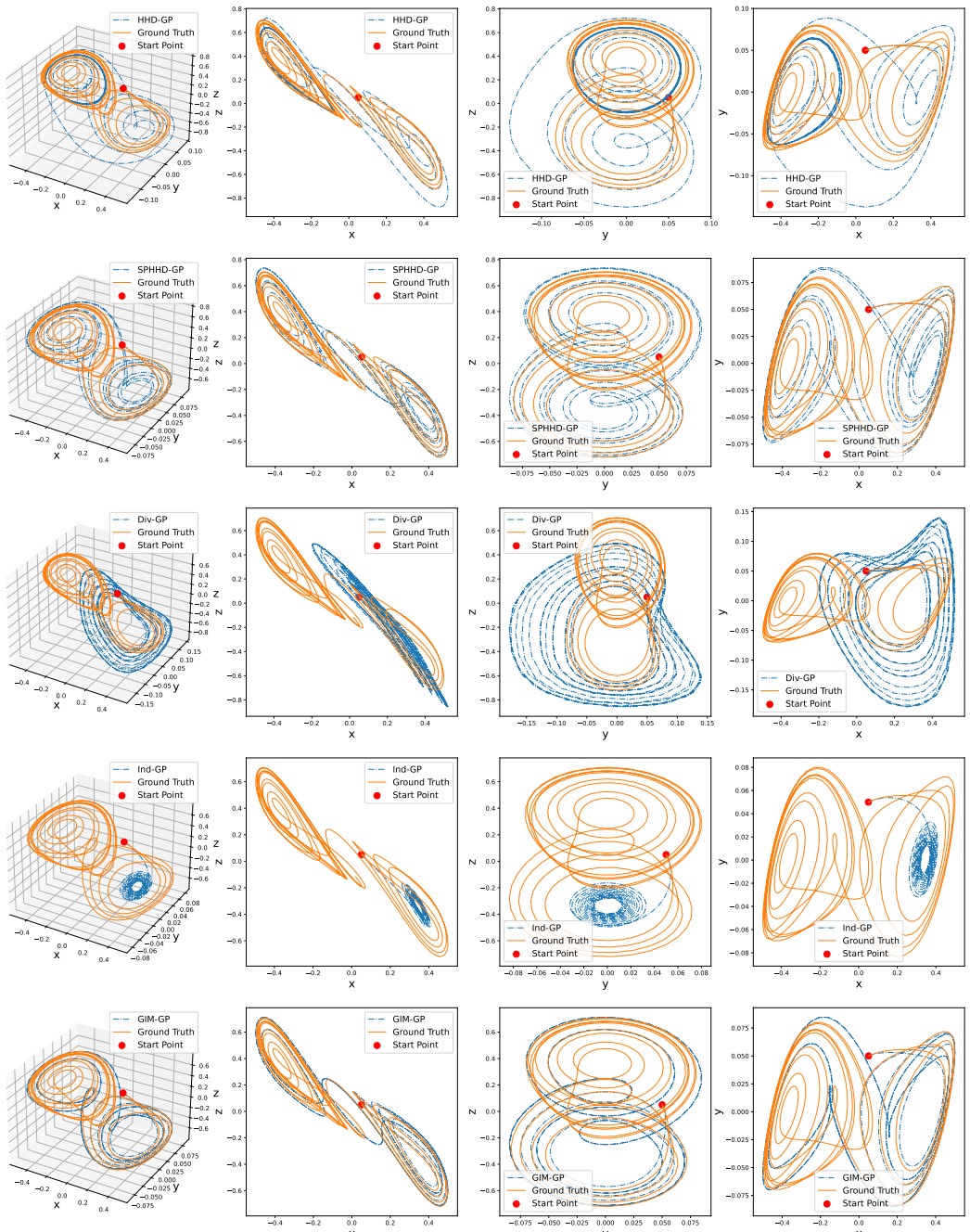

Figure 7: Comparison of predicted trajectories of the Chua circuit.

### A.8.3 RESULTS OF INCREASING NOISE LEVEL AND TRAINING DATA NUMBER

To further compare the predictive performance of the models, we evaluated them by increasing the noise and the amount of training data, respectively. In addition to the evaluation metrics used previously, including RMSE of state derivatives and energy and VPT of state trajectories, we added the mean negative log likelihood (MNLL) to evaluate the prediction uncertainty provided by the GP models, and it is defined by

$$\text{MNLL} = -\frac{1}{|\mathcal{D}|} \sum_{(\mathbf{x},\mathbf{y}) \in \mathcal{D}} \log \mathcal{N}\left(\mathbf{y} \mid \mu\left(\mathbf{x}\right), var\left(\mathbf{x}\right)\right), \tag{99}$$

where $\mathcal{D} = \{(\mathbf{x},\mathbf{y})\}$ is the test set, $\mu\left(\mathbf{x}\right)$ and $var\left(\mathbf{x}\right)$ are predicted mean and variance at a test state $\mathbf{x}$ by the GP models. The lower the MNLL, the more effectively the forecast uncertainty reflects the prediction error. Table 3 (visualized by Fig. 8) shows the experimental results on a damped pendulum with increasing standard deviation of Gaussian noise in training data, and Table 4 (visualized by Fig. 9) shows the effect of increasing the amount of training data on the performance of the model. From these results we can observe that our model (SPHHD-GP) performs best at every noise level and data amount relative to the baseline models.

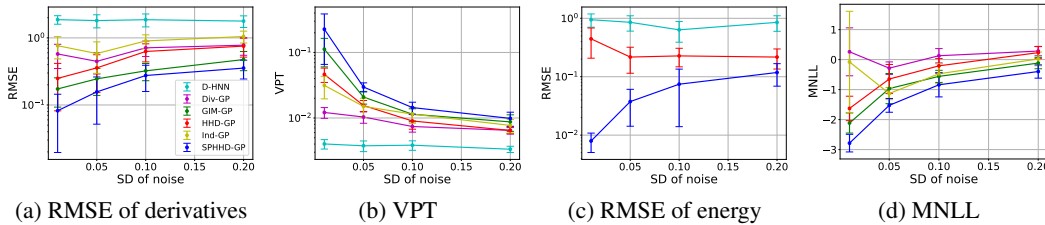

| (a) RMSE of derivatives | (b) VPT | (c) RMSE of energy | (d) MNLL |

Figure 8: Results with increasing standard deviation (SD) of Gaussian noise in training data.

Table 3: Results with increasing standard deviation $\sigma$ of Gaussian noise in training data ($\sigma = 0.01$, 0.05, 0.1 and 0.2) for the damped pendulum. The RMSEs and VPT are in the scale of $\times 10^{-2}$. All metrics are recorded in the form of mean $\pm$ standard deviation and bold font indicates best results.

| Noise SD $\sigma$ | Model | Evaluation Metrics | | | |
|---|---|---|---|---|---|
| | | RMSE of Derivatives↓ | VPT ↑ | RMSE of Energy ↓ | MNLL ↓ |
| 0.01 | D-HNN | $186.90 \pm 27.19$ | $0.40 \pm 0.07$ | $94.59 \pm 24.22$ | N/A |
| | Div-GP | $57.59 \pm 22.65$ | $1.21 \pm 0.23$ | N/A | $0.26 \pm 0.80$ |
| | Ind-GP | $76.45 \pm 27.85$ | $3.14 \pm 1.18$ | N/A | $-0.08\pm1.69$ |
| | GIM-GP | $17.34 \pm 8.05$ | $11.22 \pm 5.20$ | N/A | $-2.11 \pm 0.33$ |
| | HHD-GP (ours) | $24.87 \pm 16.75$ | $4.66 \pm 1.16$ | $44.49 \pm 23.74$ | $-1.63\pm0.41$ |
| | SPHHD-GP (ours) | $\mathbf{8.21 \pm 6.24}$ | $\mathbf{22.67 \pm 16.09}$ | $\mathbf{0.79 \pm 0.29}$ | $\mathbf{-2.78\pm0.29}$ |
| 0.05 | D-HNN | $180.99 \pm 39.94$ | $0.38 \pm 0.07$ | $85.44 \pm 25.09$ | N/A |
| | Div-GP | $44.71 \pm 15.76$ | $1.03 \pm 0.20$ | N/A | $-0.29 \pm 0.21$ |
| | Ind-GP | $58.41 \pm 28.81$ | $1.50 \pm 0.30$ | N/A | $-1.13 \pm 0.20$ |
| | GIM-GP | $24.49 \pm 10.61$ | $2.09 \pm 0.46$ | N/A | $-0.97 \pm 0.50$ |
| | HHD-GP (ours) | $35.70 \pm 20.39$ | $1.55 \pm 0.30$ | $21.72 \pm 10.26$ | $-0.65 \pm 0.49$ |
| | SPHHD-GP (ours) | $\mathbf{15.66 \pm 10.49}$ | $\mathbf{2.97 \pm 0.47}$ | $\mathbf{3.75 \pm 2.33}$ | $\mathbf{-1.53 \pm 0.23}$ |
| 0.1 | D-HNN | $186.59 \pm 40.77$ | $0.38 \pm 0.06$ | $63.49 \pm 24.51$ | N/A |
| | Div-GP | $71.07 \pm 18.38$ | $0.74 \pm 0.13$ | N/A | $0.13\pm 0.24$ |
| | Ind-GP | $90.02 \pm 21.02$ | $1.15 \pm 0.32$ | N/A | $-0.47\pm 0.33$ |
| | GIM-GP | $32.25 \pm 9.25$ | $1.14 \pm 0.24$ | N/A | $-0.56\pm 0.20$ |
| | HHD-GP (ours) | $62.70 \pm 18.84$ | $0.90 \pm 0.23$ | $22.72 \pm 7.94$ | $-0.21\pm 0.24$ |
| | SPHHD-GP (ours) | $\mathbf{27.49 \pm 11.59}$ | $\mathbf{1.44 \pm 0.30}$ | $\mathbf{7.45 \pm 6.05}$ | $\mathbf{-0.84\pm 0.40}$ |
| 0.5 | D-HNN | $177.92 \pm 35.05$ | $0.33 \pm 0.04$ | $85.26\pm 25.39$ | N/A |
| | Div-GP | $77.80 \pm 22.74$ | $0.65 \pm 0.08$ | N/A | $0.28\pm 0.15$ |
| | Ind-GP | $104.84 \pm 21.14$ | $0.77 \pm 0.18$ | N/A | $0.02\pm 0.23$ |
| | GIM-GP | $47.53 \pm 15.08$ | $0.87 \pm 0.25$ | N/A | $-0.12\pm 0.19$ |
| | HHD-GP (ours) | $75.38 \pm 23.71$ | $0.65 \pm 0.07$ | $21.70\pm 8.17$ | $0.23\pm 0.20$ |
| | SPHHD-GP (ours) | $\mathbf{35.46 \pm 11.10}$ | $\mathbf{0.99 \pm 0.24}$ | $\mathbf{11.81\pm 4.92}$ | $\mathbf{-0.40\pm 0.22}$ |

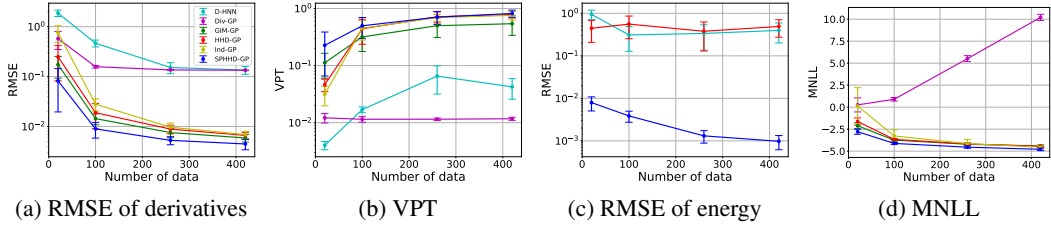

(a) RMSE of derivatives  (b) VPT  (c) RMSE of energy  (d) MNLL

Figure 9: Results with increasing number of training data.

Table 4: Results with increasing number of training data (20, 100, 260 and 420) for the damped pendulum. The RMSEs and VPT are in the scale of $\times 10^{-2}$. All metrics are recorded in the form of mean $\pm$ standard deviation and bold font indicates best results.

| Number of Training Data | Model | Evaluation Metrics | | | |
|---|---|---|---|---|---|
| | | RMSE $\downarrow$ | VPT $\uparrow$ | RMSE Energy $\downarrow$ | NMLL $\downarrow$ |
| 20 | D-HNN | $186.90 \pm 27.19$ | $0.40 \pm 0.07$ | $94.59 \pm 24.22$ | N/A |
| | Div-GP | $57.59 \pm 22.65$ | $1.21 \pm 0.23$ | N/A | $0.26 \pm 0.80$ |
| | Ind-GP | $76.45 \pm 27.85$ | $3.14 \pm 1.18$ | N/A | $-0.08 \pm 1.69$ |
| | GIM-GP | $17.34 \pm 8.05$ | $11.22 \pm 5.20$ | N/A | $-2.11 \pm 0.33$ |
| | HHD-GP (ours) | $24.87 \pm 16.75$ | $4.66 \pm 1.16$ | $44.49 \pm 23.74$ | $-1.63 \pm 0.41$ |
| | SPHHD-GP (ours) | $\mathbf{8.21 \pm 6.24}$ | $\mathbf{22.67 \pm 16.09}$ | $\mathbf{0.79 \pm 0.29}$ | $\mathbf{-2.78 \pm 0.29}$ |
| 100 | D-HNN | $46.58 \pm 7.28$ | $1.69 \pm 0.20$ | $31.02 \pm 18.24$ | N/A |
| | Div-GP | $15.78 \pm 0.93$ | $1.15 \pm 0.13$ | N/A | $0.90 \pm 0.20$ |
| | Ind-GP | $2.81 \pm 0.76$ | $45.01 \pm 16.93$ | N/A | $-3.27 \pm 0.70$ |
| | GIM-GP | $1.44 \pm 0.35$ | $31.75 \pm 14.08$ | N/A | $-3.74 \pm 0.14$ |
| | HHD-GP (ours) | $1.89 \pm 0.99$ | $44.09 \pm 20.51$ | $55.78 \pm 30.51$ | $-3.66 \pm 0.07$ |
| | SPHHD-GP (ours) | $\mathbf{0.90 \pm 0.31}$ | $\mathbf{49.72 \pm 19.75}$ | $\mathbf{0.39 \pm 0.11}$ | $\mathbf{-4.12 \pm 0.11}$ |
| 260 | D-HNN | $15.21 \pm 3.77$ | $6.55 \pm 3.37$ | $33.63 \pm 20.02$ | N/A |
| | Div-GP | $13.60 \pm 0.17$ | $1.15 \pm 0.05$ | N/A | $5.51 \pm 0.34$ |
| | Ind-GP | $0.97 \pm 0.20$ | $69.18 \pm 9.79$ | N/A | $-4.14 \pm 0.48$ |
| | GIM-GP | $0.76 \pm 0.16$ | $49.95 \pm 18.89$ | N/A | $-4.22 \pm 0.13$ |
| | HHD-GP (ours) | $0.89 \pm 0.17$ | $70.55 \pm 18.08$ | $37.83 \pm 24.81$ | $-4.18 \pm 0.10$ |
| | SPHHD-GP (ours) | $\mathbf{0.53 \pm 0.10}$ | $\mathbf{71.62 \pm 15.66}$ | $\mathbf{0.13 \pm 0.04}$ | $\mathbf{-4.54 \pm 0.12}$ |
| 420 | D-HNN | $13.48 \pm 2.37$ | $4.24 \pm 1.67$ | $39.69 \pm 19.62$ | N/A |
| | Div-GP | $13.39 \pm 0.10$ | $1.17 \pm 0.07$ | N/A | $10.21 \pm 0.35$ |
| | Ind-GP | $0.69 \pm 0.11$ | $75.71 \pm 12.58$ | N/A | $-4.56 \pm 0.20$ |
| | GIM-GP | $0.59 \pm 0.10$ | $54.11 \pm 20.25$ | N/A | $-4.43 \pm 0.13$ |
| | HHD-GP (ours) | $0.66 \pm 0.10$ | $78.55 \pm 10.27$ | $49.03 \pm 21.57$ | $-4.44 \pm 0.10$ |
| | SPHHD-GP (ours) | $\mathbf{0.45 \pm 0.10}$ | $\mathbf{81.74 \pm 11.78}$ | $\mathbf{0.10 \pm 0.04}$ | $\mathbf{-4.79 \pm 0.15}$ |

### A.8.4 EXPERIMENTS OF PREDICTING DYNAMICS WITH UNSEEN FRICTIONS

To demonstrate the interpretability of the learned curl-free features, we adapted the trained model to predict dynamics with different friction coefficients. As detailed in the Appendix A.1, the curl-free dynamics in the system is always caused by dissipative forces in the system (*e.g.*, friction in the mass-spring system and the pendulum). Utilizing this interpretation and the additive structure of the HHD-based models, we can generalize the models to perform inference over dynamics with different friction coefficients. We performed this experiment by first training HHD-GP, SPHHD-GP and D-HNN with data when $\rho = 0.1$ ($\rho$ is the friction coefficients), then generalizing the trained models to dynamics when $\rho = 0.05$ and $\rho = 0.5$ by multiplying the learned curl-free component with constants $0.5$ and $5.0$, respectively. The results are shown in Fig. 10, where we can observe that the three models accurately predict the trajectories of the systems with a friction coefficient of $0.1$ (first column in Fig. 10), but HHD-GP and D-HNN are difficult to generalize to cases with friction coefficients of 0.05 and 0.5 (second and third columns in Fig. 10). In contrast, SPHHD-GP effectively captures the dynamics of different friction conditions, even though it has not been trained for these specific friction coefficients. This adaptability is valuable because it allows the model to be applied to real-world scenarios where friction coefficients may change.

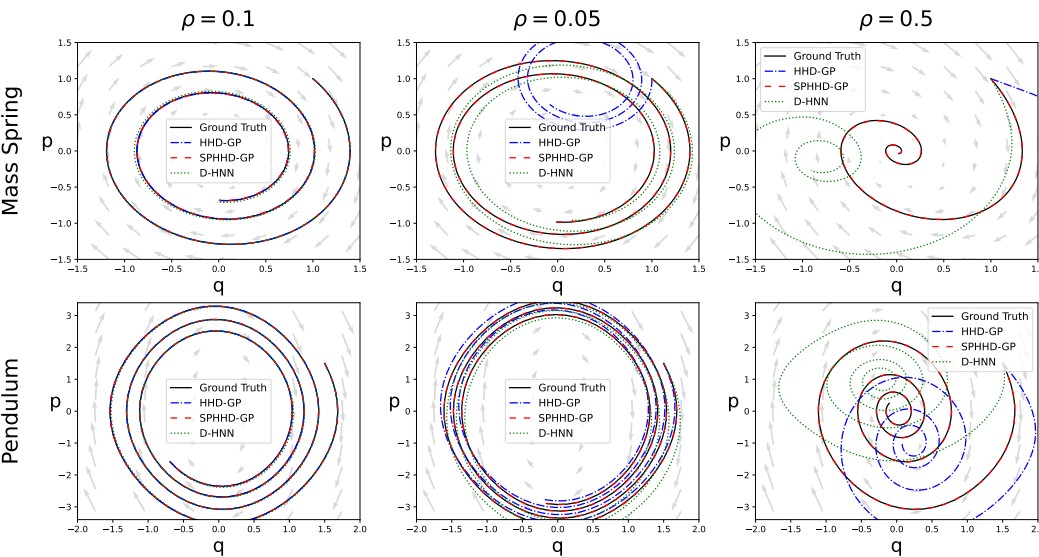

Figure 10: Adapting the trained models to predict trajectories for different friction coefficients. The models are trained under $\rho = 0.1$, where $\rho$ is the friction coefficients, then they are generalized to predict trajectories for $\rho = 0.05$ and $\rho = 0.5$ by multiplying the learned curl-free component with constants $0.5$ and $5.0$, respectively. The lines in the figure represent predicted trajectories from different models (distinguished by different line styles), with the initial points for the two systems (the first and second rows) being $(1, 1)$ and $(1.5, 1.5)$, respectively. The background vector fields are the ground truth dynamics.

