# OpenReview forum: "Learning Dynamical Systems with Helmholtz-Hodge Decomposition and Gaussian Processes"
_ICLR.cc/2024/Conference — Submitted to ICLR 2024_

### Official Review · Reviewer_e4ns · 2023-10-27

**Soundness:** 4 excellent
**Presentation:** 4 excellent
**Contribution:** 4 excellent
**Rating:** 8
**Confidence:** 3

**Summary:**

The authors propose to learn from dynamical data by considering the Helmholtz-Hodge decomposition and modeling separate GPs to the divergence-free and the curl-free components of the system. To enable identifiability for the resulting additive model, Euclidean symmetries are imposed to the components. The obtained experimental results are promising when compared to a neural network model and other GP-based methods.

**Strengths:**

Learning dynamical systems is a fundamental problem that has been covered by the machine learning community from many angles. By considering the Helmholtz-Hodge decomposition and directly including symmetry constraints in the kernel function, the authors present a very interesting and relevant contribution.

The overall presentation is laid very carefully and the text is well written. The authors take their time to explain all the important concepts and related work in the main text, which greatly aids the reader. Despite tackling several concepts of dynamical systems and vector calculus, I found the manuscript to be very didactic and easy to follow.

Theoretical and implementation details are presented in the the very comprehensive appendix, which also includes additional plots from the experiments.

**Weaknesses:**

I believe it would be of interest to include at least one scenario where the number of training observations is larger to verify if the SPHHD-GP maintains its high gains compared to the baselines. I think a larger Gaussian noise could also be tested, since a standard deviation of 0.01 (0.0001 variance) seems a bit too low.

The authors consider only the RMSE and VPT metrics in the experiments. The predicted variances provided by the GP models should also be included in the evaluation, e.g., by computing the log predictive density.

**Questions:**

It is difficult to visually differentiate the colors of the curves in Figs. 5, 6, 7. Is it possible to consider distinct line styles, similar to Fig. 2?

---

> ### Author Response · Authors · 2023-11-22
> **Response to Reviewer e4ns**
>
> We thank the reviewer for the constructive comments. We are glad that you believe that this work can bring a valuable contribution to the community. We have carefully gone through your comments, so we have done additional experiments and revised our paper accordingly. We believe addressing these points in the manuscript would indeed make the paper better.
>
> ---
>
> > **1.Comment**: I believe it would be of interest to include at least one scenario where the number of training observations is larger to verify if the SPHHD-GP maintains its high gains compared to the baselines. I think a larger Gaussian noise could also be tested, since a standard deviation of 0.01 (0.0001 variance) seems a bit too low.
>
>
> **Response**: Thanks to the reviewer for this suggestion. We agree that a more thorough validation will strengthen our paper. So we added experiments of evaluating the models performance under various noise and number of training data in the Appendix A.8.3 of the updated draft (colored by blue). Specifically, the first part of additional experiments were perfomed on a damped pendulum with increasing standard deviation (0.01, 0.05, 0.10 and 0.20) of Gaussian noise in training data. The results in Fig.8 and Table 3 shows that our model (SPHHD-GP) performs best at every noise level relative to the baseline models, meaning that our model is more robust to data noise. And the second part of experiments investigated the effect of the number of training data (20, 100, 260, 420) on the model performance. The results are shown in Fig.9 and Table 4, where we can observe that the performance of the models are closer as the training data increases, but our model still shows the best performance on any number of training data. Furthermore, an important observation from Fig.8(c) and Fig.9(c) is that both HHD-GP and D-HNN consistently generate significant errors in predicting energy evolution, and the errors are almost unaffected by the noise and quantity of training data, meaning these two models cannot learn the correct decomposition under any training data conditions. In contrast, the energy errors generated by SPHHD-GP exhibit a increasing trend with increasing noise and an decreasing trend with increasing number of training data. This observation further proves the importance and effectiveness of addressing model identifiability through symmetry constraints.
>
> ---
>
> > **2.Comment**: The authors consider only the RMSE and VPT metrics in the experiments. The predicted variances provided by the GP models should also be included in the evaluation, e.g., by computing the log predictive density.
>
> **Response**: Providing uncertainty estimates about predictions is one of the unique advantages of the GP model, so we fully agree with the reviewer's suggestion to add evaluation metrics in this aspect, so we added the mean negative log likelihood (MNLL) as one of our measures when evaluating the models performance under different noise and number of training data. The definition of MNLL and the corresponding experimental results can be found in Appendix A.8.3 of the updated draft (colored by blue). The lower the MNLL, the more effectively the forecast uncertainty reflects the prediction error. As expected, MNLLs of the GP models exhibit a increasing trend with increasing noise and an decreasing trend with increasing number. An exception is Div-GP (divergence-free GP, a GP model that can only learn conservative dynamics), which produces MNLL with a different trend compared to other GP models. This is because it uses incorrect prior knowledge as an inductive bias, so the uncertainty estimates it produces cannot serve as a reference for prediction errors. And our model (SPHHD-GP) has the lowest MNLL for every noise and number of training data, meaning that the uncertainty estimates produced by our model can better reflect the prediction error.
>
> ---
>
> > **3.Comment**: It is difficult to visually differentiate the colors of the curves in Figs. 5, 6, 7. Is it possible to consider distinct line styles, similar to Fig. 2?
>
> **Response**: We thank the reviewer for pointing this out. We now used distinct line styles to represent trajectories generated by different models.
>
> ---
> Thank you again for the constructive comments. We hope these explanations resolve the concerns. Any further questions or suggestions would be greatly appreciated.

---

> > ### Comment · Reviewer_e4ns · 2023-11-22
> > **Thanks for the update**
> >
> > I would like to thank the authors for the responses and the several updates to the manuscript. I will maintain my original rating.

---

### Official Review · Reviewer_oXvz · 2023-11-01

**Soundness:** 3 good
**Presentation:** 3 good
**Contribution:** 2 fair
**Rating:** 6
**Confidence:** 3

**Summary:**

The paper proposes a dynamical system based on the Helmholtz–Hodge decomposition (HHD) and Gaussian processes (GPs). The authors demonstrate how to learn the dynamic systems in the HHD form where each component is modeled as a GP.

**Strengths:**

1. The authors propose to decompose the dynamical system as curl-free and divergence-free components and learn the decomposition using GPs. The formulation is interesting and has physically-meaningful interpretation.
2. To address identifiability, the authors incorporate symmetry constraints on the HHD components, and provide theoretical characterizations of the resulting kernels.
3. Empirical results demonstrate that the proposed method achieves improved accuracy in modeling ODEs as well as energy evolution of the systems.

**Weaknesses:**

1. Computational complexity for the method could be prohibitive for high-dimensional problems where the partial derivatives wrt. the input need to be computed. Low-rank structures are typically used in standard dynamical systems, but it is not leveraged in the proposed approach.
2.  As suggested by the authors, the decomposition is not unique due to the unidentifiability of the harmony component. The authors address this issue by imposing symmetry constraints; however, the uniqueness and characteristics of the resulting decomposition are not investigated.
3. Another concern is the novelty of the approach compared to the multiple kernel learning literature. The proposed method appears to be an instance of learning a combination of two pre-defined kernels.

**Questions:**

1. What is the computational complexity of the method in terms of the sample size and input dimension?
2. How does the method differ from multiple kernel learning with two given kernels?

---

> ### Author Response · Authors · 2023-11-22
> **Response to Reviewer oXvz (Part-1 out of 2)**
>
> We thank the reviewer for the constructive comments and for taking the time to thoroughly read this paper. We have carefully gone through your comments and addressed each of them point-by-point. Sorry for the late response.
>
> ---
>
> > **1.Comment (W1&Q1)**: Computational complexity for the method could be prohibitive for high-dimensional problems where the partial derivatives wrt. the input need to be computed. Low-rank structures are typically used in standard dynamical systems, but it is not leveraged in the proposed approach. What is the computational complexity of the method in terms of the sample size and input dimension?
>
>
> **Response**: Thanks to the reviewer for raising this concern. Computational complexity is indeed one of the main drawbacks of GPs. Define $m$ as the sample size and $n$ as the number of the input dimension, then the time complexity of our method is dominated by the Cholesky decomposition to invert the $mn \times mn$ covariance matrix, which takes $\left ( m n \right )^3/6$ operations, and all other operations are at most quadratic in $m$ and $n$. So the time complexity of our model is $\mathcal{O} \left ( m^3 n^3 \right )$. We understand the reviewer's concern about computational complexity, but at the expense of this computational cost, our model achieves high data efficiency and inherent regularization. We thank the reviewer for your inspiration in leveraging low-rank structures in dynamical systems to approximate the covariance matrix, and we are very intrigued about extensions of our work in reducing the computational complexity in future work. And we now discussed this limitation of computational complexity in the section of Conlusion in the updated draft (colored by blue).
>
> ---
>
> > **2.Comment (W2)**:  As suggested by the authors, the decomposition is not unique due to the unidentifiability of the harmony component. The authors address this issue by imposing symmetry constraints; however, the uniqueness and characteristics of the resulting decomposition are not investigated.
>
> **Response**:
>
> We thank the reviewer for giving us the opportunity to clarify the the uniqueness of the resulting decomposition. A unique decomposition makes our model identifiable. To show the identifiability of our model, an additional experiment on the damped pendulum system was perfomed to demonstrate the ***consistency*** of our model. Consistency is a key property of identifiable models [C1, C2]. If the model is identifiable, then the parameter estimates should converge to the true values as the amount of data increases. Therefore, we added experiments of evaluating the models performance with increasing number of training data (n=20, 100, 260, 420) in the Appendix A.8.3 of the updated draft (colored by blue). We invite the reviewer to review Fig.9(c) and Table 4, which shows that models without symmetry constraints (HHD-GP and D-HNN) consistently generate significant errors in predicting energy evolution, and the errors are almost unaffected by the number of training data, meaning that the model parameters cannot converge as the amount of data increases. In contrast, the energy errors of SPHHD-GP (Symmetry-preserving HHD-GP) exhibit a decreasing trend with increasing number of training data. The experimental results proves the consistency of our model, but since consistency is only a necessary condition for identifiability, we can only conclude that symmetry constraints improves the model identifiability, i.e. reducing the space of harmonic components. To prove that the existence of harmonic components are completely eliminated by the symmetry constraints, we added a theoretical verification that for the three dynamical systems in our experiments, the non-uniqueness of HHD has been solved through forced symmetries. We invite the reviewer to review this analysis in Appendix A.6 (colored by blue) of the updated draft.
>
> We would like to further clarify that although we currently do not have a theoretical result to characterize the conditions for symmetry to effectively eliminate harmonic components in HHD, we have demonstrated the potential of symmetry in enhancing model identifiability, an area that has been relatively unexplored before. Current research mainly focuses on using symmetry to improve the predictive power of models. Therefore, we believe that studying the effect of symmetry on model identifiability is a very valuable future direction, which is important for model interpretability.
>
> ---

---

> > ### Author Response · Authors · 2023-11-22
> > **Response to Reviewer oXvz (Part-2 out of 2)**
> >
> > ---
> >
> > > **3.Comment (W3&Q2)**: Another concern is the novelty of the approach compared to the multiple kernel learning literature. The proposed method appears to be an instance of learning a combination of two pre-defined kernels. How does the method differ from multiple kernel learning with two given kernels?
> >
> > **Response**: We appreciate the reviewer for raising this concern. We would like to clarify the difference between our method and multiple kernel learning (MKL) in terms of their underlying principles and objectives. MKL focuses on learning the optimal combination of predefined kernels [C3], while our model assumes an additive structure in the data and model it by combining independent kernels in an additive manner. This additive structure in dynamical systems is guaranteed by the Helmholtz-Hodge decomposition (HHD), which states that a dynamical system can be decomposed into a sum of curl-free and div-free components. Therefore, we would like to emphasize that our main contribution lies in the contruction of divergence-free kernels and curl-free kernels, as well as their symmetry-preserving extensions,  which not only improve the prediction performance of the model but also address the identifiability problem caused by harmonic components in additive structures.
> >
> > ---
> > Thank you again for the constructive comments. We hope these explanations resolve the concerns. Any further questions or suggestions would be greatly appreciated.
> >
> > ## References
> >
> > [C1] Gabrielsen, Arne. *Consistency and identifiability*. Journal of Econometrics 8.2 (1978): 261-263.
> >
> > [C2] Ljung, Lennart. *On consistency and identifiability*. Stochastic Systems: Modeling, Identification and Optimization, I. Berlin, Heidelberg: Springer Berlin Heidelberg, 2009. 169-190.
> >
> > [C3] Gönen, Mehmet, and Ethem Alpaydın. *Multiple kernel learning algorithms*. The Journal of Machine Learning Research 12 (2011): 2211-2268.

---

> > > ### Comment · Reviewer_oXvz · 2023-12-03
> > > **After rebuttal**
> > >
> > > I'd like to thank the authors for addressing my comments. The computational complexity O(m^3 n^3) is indeed very high compared to a standard GP (O(m^3)), and I understand that this may be necessary for the specific GP structure proposed. I appreciate that the authors can clearly stated the computational complexity in the revised paper. Thus, I raised my score accordingly.

---

### Official Review · Reviewer_JNyE · 2023-11-01

**Soundness:** 3 good
**Presentation:** 3 good
**Contribution:** 3 good
**Rating:** 6
**Confidence:** 4

**Summary:**

This paper presents a method to learning complex systems. The method has three components: using Helmholtz-Hodge decomposition on the phase plane of the dynamic systems, as influenced by Greydanus et al 2019 (Hamiltonian NN), to decompose system dynamics into curl-free and div-free components; using Gaussian Process as prior to capture those two components separately from the training data; and use Euclidean group to further enforce the symmetries of GP models.

**Strengths:**

This paper addresses an important problem in dynamics prediction. Inspired by D-HNN (2022), the method is novel, in particular the usage of Euclidean Group to further enforce the symmetries. There is sufficient amount of empirical evidence to back up the claims. The paper contains extensive amount of theoretical work, and is written reasonably well.

**Weaknesses:**

The experimental results section needs some more clarification, see the questions in the next sections.

The non-uniqueness of HHD is well-known. The authors use it as in section 3.3 to introduce the enforcement of the symmetries of the GP model. However it's unclear that the enforcement of the symmetries can completely eliminate the non-uniqueness of HHD. (My understanding is negative). Although the preservation of symmetries is usually a desirable property to model dynamics systems, either a theoretical guarantee or empirical validation that the non-uniqueness can be addressed or mitigated is highly desirable.

Real data are never clean. The authors added noise into the training data. But my concern is that the noise magnitude is too small (0.01, only 1% of the data range. I would like to see how well the method behaves when larger magnitude of noise is added, for instance, with std. dev. of 0.01, 0.05, 0.10 and 0.20.

**Questions:**

1. In Table 1, what are the std. dev. values collected from? My understanding is that random initial values are used in the GP model. Are the randomness of the initial values cause the derivations?

2. I am trying to understand the reported RMSE and VPT in Table 1. For each system, 20 pairs of data are generated for training, where are the testing data generated? In the temporal sense, are the testing data lie in the future of the training data, or the training and testing data pairs are intermingled with respect of time?

---

> ### Author Response · Authors · 2023-11-22
> **Response to Reviewer JNyE (Part-1 out of 2)**
>
> We would like to thank the reviewer for the thoughtful comments and suggestions, which have helped to improve the quality of our manuscript.
>
> ---
>
> > **1. Comment**: The non-uniqueness of HHD is well-known. The authors use it as in section 3.3 to introduce the enforcement of the symmetries of the GP model. However it's unclear that the enforcement of the symmetries can completely eliminate the non-uniqueness of HHD. (My understanding is negative). Although the preservation of symmetries is usually a desirable property to model dynamics systems, either a theoretical guarantee or empirical validation that the non-uniqueness can be addressed or mitigated is highly desirable.
>
> **Response**: We thank the reviewer for giving us the opportunity to present a theoretical analysis that for the three dynamical systems in our experiments the non-uniqueness of HHD is solved through forced symmetries. The non-uniqueness of HHD is caused by the presence of harmonic components (vector fields that are both curl-free and divergence-free). To make HHD unique, we propose to imposing symmetry constraints to the space of divergence-free vector fields and the space of curl-free vector fields respectively, with the corresponding symmetry groups defined as $G_{div}$ and $G_{curl}$. So the HHD will be unique if there are no harmonic vector fields respecting the union of $G_{div}$ and $G_{curl}$.
>
> A harmonic vector field on $\mathbb{R}^n$ can be represented by the gradient field of a harmonic function $h$, which is a scalar function satisfying $\nabla \cdot \nabla h = 0$ (Laplace's equation). For the three dynamical systems used in our experiments, it can be easily verified that the existence of $h$ can be eliminated through forced symmetries. The symmetry group union of a damped mass-spring system consists of a rotation group $G_{div} = SO \left ( 2 \right )$ and a translation group $G_{curl} = \left \\{ \left ( g, 0 \right ) \mid g \in \mathbb{R} \right \\}$. A harmonic vector field $\mathbf{f} = \nabla h$ that satisfies this translation symmetry implies that its harmonic function $h \left ( q, p \right )$ is independent of the variable $q$, so the harmonic vector field is given by $\mathbf{f} = \left (0, \partial_p h \right )$ and $\partial_p h$ should be a constant to satisfy the Laplace’s equation, but the harmonic vector field in the form of constant clearly contradicts rotation symmetry $G_{div}$. Therefore, there is no harmonic vector field that respects both the symmetry groups $G_{div}$ and $G_{curl}$. Similar conclusions can be drawn for the damped pendulum and the Chua circuit. The Laplace's equation and translation symmetry imply that harmonic vector fields can only exist in the form of constant vector fields. However, constant vector fields obviously contradict odd symmetry or mirror symmetry. We added this theoretical analysis to the Section A.6 in the appendix of the updated draft.
>
> Although we currently do not have a theoretical result to characterize the conditions for symmetry to effectively eliminate harmonic components in HHD, we have demonstrated the potential of symmetry in enhancing model identifiability, an area that has been relatively unexplored before. Current research mainly focuses on using symmetry to improve the predictive power of models. Therefore, we believe that studying the effect of symmetry on model identifiability is a very valuable future direction, which is important for model interpretability.
>
> ---
>
> > **2. Comment**: Real data are never clean. The authors added noise into the training data. But my concern is that the noise magnitude is too small (0.01, only 1\% of the data range. I would like to see how well the method behaves when larger magnitude of noise is added, for instance, with std. dev. of 0.01, 0.05, 0.10 and 0.20.
>
> **Response**: Thank the reviewer for this valuable suggestions. We added the experiments of evaluating the models performance with increasing standard deviation (0.01, 0.05, 0.10 and 0.20) of Gaussian noise in training data. We find that our model (SPHHD-GP) performs best at every noise level relative to the baseline models, meaning that our model is more robust to data noise. We invite the reviewer to Fig.8 and Table 3 in the updated draft to check the experimental results. In addition, according to the suggestion of Reviewer e4ns, we added the experiments of investigating the effect of the number of training data on the model performance. The results are shown in Fig.9 and Table 4, where we can observe that our model also shows the best performance on any number of training data.
>
> ---

---

> > ### Author Response · Authors · 2023-11-22
> > **Response to Reviewer JNyE (Part-2 out of 2)**
> >
> > > **3. Comment**: In Table 1, what are the std. dev. values collected from? My understanding is that random initial values are used in the GP model. Are the randomness of the initial values cause the derivations?
> >
> > **Response**: In order to mitigate the influence of the distribution of training data and the initial parameters of the model on the experimental results, all experimental results are based on 10 independent experiments conducted by resampling the training set and the initial parameters of the model. Sorry for causing this confusion, we now highlight this setting in the caption of Table 1 (colored by blue).
> >
> > ---
> > > **4. Comment**: I am trying to understand the reported RMSE and VPT in Table 1. For each system, 20 pairs of data are generated for training, where are the testing data generated? In the temporal sense, are the testing data lie in the future of the training data, or the training and testing data pairs are intermingled with respect of time?
> >
> > **Response**: Sorry for causing this confusion. In the original draft, the generation of test data in placed in the Appendix. To address this confusion, we now included the generation of test data in the first paragraph of Section 6 in the updated draft (colored by blue). For the RMSE, the test set (state-derivative pairs) was generated by sampling a grid with a resolution of 10 points along each dimension of the dynamical systems. And for the VPT, the test set consists of 50 trajectories simulated from randomly sampled initial states and each trajectory was generated by integrating an initial state forward in time at a frequency of 25 Hz for 15 seconds. There is no connection between the sampling of training data and test data.
> >
> > ---
> > Thank you again for the constructive comments. We hope these explanations resolve the concerns. Any further questions or suggestions would be greatly appreciated.

---

### Official Review · Reviewer_uHv2 · 2023-11-05

**Soundness:** 2 fair
**Presentation:** 2 fair
**Contribution:** 2 fair
**Rating:** 3
**Confidence:** 4

**Summary:**

In this work, authors present a Gaussian process model that decomposes the dynamics of a dissipative system into the curl-free and divergence-free terms. Further, these terms are learned from the noisy ground truth data. The framework is claimed to have the additional advantage of being interpretable due to the separate learning of the two terms in the dynamics. Empirical studies on damped mass-spring system, damped pendulum, and Chua circuit shows superior performance over the baselines.

**Strengths:**

The main strengths of the paper are as follows.

S1. Presents a GP-based framework that can inherently handle noisy data. This is in contrast to most of the works in the literature that employs neural-based approaches.

S2. Decomposes the dynamics into curl-free and div-free terms. This allows learning the non-conservative and conservative components of the dynamics separately.

S3. Empirically demonstrates that the presented framework is superior to the baselines on damped spring systems, damped pendulum and Chua circuit.

**Weaknesses:**

There are several weaknesses for the paper.

W1. There are several works in the literature which demonstrates how the Lagrangian, (port-)Hamiltonian, and neural ODE based NN frameworks can be used to model dissipative dynamical systems. Authors have not given any mention of such frameworks in the introduction, which provides a feeling that there is no prior work in this area. The reference comes much later when discussing the baselines. This should be clearly included in the introduction. Some relevant works are as follows.
* Desai, S.A., Mattheakis, M., Sondak, D., Protopapas, P. and Roberts, S.J., 2021. Port-Hamiltonian neural networks for learning explicit time-dependent dynamical systems. Physical Review E, 104(3), p.034312.
* Sosanya, A. and Greydanus, S., 2022. Dissipative hamiltonian neural networks: Learning dissipative and conservative dynamics separately. arXiv preprint arXiv:2201.10085.
* Drgoňa, J., Tuor, A., Vasisht, S. and Vrabie, D., 2022. Dissipative deep neural dynamical systems. IEEE Open Journal of Control Systems, 1, pp.100-112.
* Gruver, N., Finzi, M., Stanton, S. and Wilson, A.G., 2022. Deconstructing the inductive biases of hamiltonian neural networks. arXiv preprint arXiv:2202.04836.
* Bhattoo, R., Ranu, S. and Krishnan, N.A., 2023. Learning the dynamics of particle-based systems with Lagrangian graph neural networks. Machine Learning: Science and Technology, 4(1), p.015003.

W2. The claim on the interpretability presented in the abstract is not substantiated later on in the empirical experiments or results. It is not clear how the framework is interpretable especially for a system with larger number of degrees of freedom.

W3. The experiments performed are on very (very) simple systems such as a damped spring and damped pendulum. The community has moved forward from these experiments. Please see the experiments in the references mentioned against W1. Especially, the experiments on simple one-degree of freedom toy examples are not enough to show the applicability of the approach to any realistic problems. For this some demonstration on larger systems with more degrees of freedom (~50-100) should be conducted. Note that realistic systems can have much higher degrees of freedom.

W4. Baselines are not appropriately chosen. Again, references provided in W1 should be used for baselines. Some of the baselines that can be included are graph neural ODE, Lagrangian and Hamiltonian NN with dissipative terms, Lagrangian and Hamiltonian graph NNs with the dissipative terms to name a few.

**Questions:**

In continuation to the weaknesses, the following questions/comments need to be addressed.

Q1. The evaluation metrics are important. This should be preferably included in the main manuscript and not the appendix. Moreover, why are other metrics such as energy error, momentum error etc. not included? This allows meaningful interpretation of the error in the learned dynamics.

Q2. It is not clear what the authors mean by the claim that the framework is interpretable. Do they mean that the dissipative and non-dissipative terms are learned separately? This is not necessarily interpretable. There are interesting works on interpretability. For instance, see:
* Cranmer, M., Sanchez Gonzalez, A., Battaglia, P., Xu, R., Cranmer, K., Spergel, D. and Ho, S., 2020. Discovering symbolic models from deep learning with inductive biases. Advances in Neural Information Processing Systems, 33, pp.17429-17442.
* Cranmer, M.D., Xu, R., Battaglia, P. and Ho, S., 2019. Learning symbolic physics with graph networks. arXiv preprint arXiv:1909.05862.

Q3. How does the system perform on more complex systems such as 50-mass spring systems or 5-pendulum systems. GPs are known to have issues with larger input features. Can the present approach be extended to such realistic systems?

Q4. It is not clear how the input features will be employed for the present approach in a multi-degree of freedom system. Specifically, whether the approach is permutation invariant or not is not clear. That is, does the order in which the degrees of freedom are provided as the input matter or not? Authors should clarify.

---

> ### Author Response · Authors · 2023-11-22
> **Response to Reviewer uHv2 (Part-1 out of 3)**
>
> We thank the reviewer’s detailed and constructive comments. We have carefully gone through the comments and below are our responses point-by-point. Sorry for our late response.
>
> ---
>
> > **1. Comment (W1)**: There are several works in the literature used to model dissipative dynamical systems, which should be clearly included in the introduction [1-5].
>
> **Response**: We thank the reviewer for the suggestion and for providing these reference methods. We now include these works in the Introduction of the updated draft (colored by blue). We illustrate the essential differences between our approach and these methods. We emphasize that our focus is on designing an additive GP model whose each component satisfies certain differential invariants (either free of divergence or of curl), rather than constructing ML models according to some physically governing equations (e.g. Hamiltonian or Lagrangian equations). Therefore, our model has a wider scope of applicability. For example, in our experiments, the methods mentioned by the reviewer are difficult to apply to the Chua circuit system, which is a 3D chaotic system that is hard to describe using Hamiltonian or Lagrangian equations.
>
> ---
>
> > **2. Comment (W2 & Q2)**: The claim on the interpretability presented in the abstract is not substantiated later on in the empirical experiments or results. It is not clear what the authors mean by the claim that the framework is interpretable. Do they mean that the dissipative and non-dissipative terms are learned separately? This is not necessarily interpretable. There are interesting works on interpretability [5,6].
>
> **Response**: Thanks to the reviewer for recommending two such interesting works. Indeed, these deep symbolic regression methods are known to have high interpretability, because they can recover symbolic expressions of dynamical systems from learned models. Your comment prompt us to further clarify and demonstrate interpretability of our model. While our model cannot extract explicit analytical forms, it instead provides a unique and valuable interpretation by decomposing the underlying system into a divergence-free component and a curl-free component, which represent the distinct physical phenomena at play within the system. The current work emphasizes the connection between HHD and the generalized Hamiltonian formalism. As described in Appendix A, our model can learn a generalized Hamiltonian function for any dimensional system, as long as the correct form of HHD is learned. To demonstrate this interpretability, the experiments detailed in Section 6.2 shows that the div-free features learned by the model with symmetry constraints (SPHHD-GP) can be used to accurately predict the energy evolution of dynamical systems. In contrast, models without symmetry constraints (HHD-GP and D-HNN) cannot learn the correct HHD decomposition due to model identifiability issues, so the features they learn are not physically interpretable. To further show the interpretability of our model, we add experiments of generalizing the learned models to predict dynamics with unseen friction coefficients, which demonstrate the interpretability of learned curl-free features. We invite the reviewer to view the experimental results in the Appendix A.8.4 (colored by blue) of the updated draft. We state that the interpretability of our model is guaranteed by the identifiability of the model achieved by symmetry constraints. Therefore, in addition to these experimental results, we have now added theoretical verification (Appendix A.6 in the updated draft) for the three demonstration examples that the non-identifiability of our model (i.e. non-uniqueness of the HHD) is solved through forced symmetries.

---

> > ### Author Response · Authors · 2023-11-22
> > **Response to Reviewer uHv2 (Part-2 out of 3)**
> >
> > ---
> > > **3. Comment (W3 & Q3)**: The experiments performed are on very (very) simple systems such as a damped spring and damped pendulum. The community has moved forward from these experiments. Please see the experiments in the references mentioned against W1. Especially, the experiments on simple one-degree of freedom toy examples are not enough to show the applicability of the approach to any realistic problems. For this some demonstration on larger systems with more degrees of freedom (~50-100) should be conducted. Note that realistic systems can have much higher degrees of freedom. How does the system perform on more complex systems such as 50-mass spring systems or 5-pendulum systems. GPs are known to have issues with larger input features. Can the present approach be extended to such realistic systems?
> >
> > **Response**: Although there are now several works dedicated to modeling multi-object dynamical systems, two-dimensional systems including mass springs, pendulums, etc. are still among the main benchmark systems for studying and testing data-driven dynamic models, as in the references [1,2,3,4] provided by the reviewer in W1 and some more recent representative works [B1, B2, B3]. And in addition to the damped mass-spring system and the damped pendulum, our model has also been tested on a three-dimensional chaotic system, which is generally regarded as a system with complex behavior in existing works. However, we fully agree with the reviewer's concern about computational complexity of the proposed model, and this is indeed the most significant drawback of GP models. Despite the high computational complexity, GP models have several distinctive advantages, such as high data efficiency and inherent regularization, which have been verified in our experiments. Therefore, we believe that the proposed model has potential applications for many realistic systems, especially those where only sparse noise observations are available. We now add a discussion of computational complexity limitations in the section of Conclusion (Section.7 in the updated draft). To reduce the computational complexity, combining the proposed model with methods such as sparse variational inference is a fruitful future direction of inquiry.
> >
> > ---
> > > **4. Comment (W4)**: Baselines are not appropriately chosen. Again, references provided in W1 should be used for baselines. Some of the baselines that can be included are graph neural ODE, Lagrangian and Hamiltonian NN with dissipative terms, Lagrangian and Hamiltonian graph NNs with the dissipative terms to name a few.
> >
> > **Response**: Thanks again to the reviewer for providing these reference methods. In our response to W1, we described the essential difference between our model and these approaches. Moreover, considering the distinctive characteristics of GPs, we have chosen more GP-based models as baseline models. However, we completely agree with the reviewer that physical equations-governed NNs should be incorporated into the comparisons. Therefore, the submitted draft included D-HNN (Dissipative Hamiltonian Neural Network, [2] in W1) as one of the baseline models. As expected, it tends to overfit noisy data compared to nonparametric models. At the same time, the lack of consideration of model identifiability prevents the D-HNN results from providing a physically plausible explanation. We believe that this observation is sufficient to emphasize the advantages of our model compared to physical equations-governed NNs.
> >
> > ---
> > > **5. Comment (Q1)**: The evaluation metrics are important. This should be preferably included in the main manuscript and not the appendix. Moreover, why are other metrics such as energy error, momentum error etc. not included? This allows meaningful interpretation of the error in the learned dynamics.
> >
> > **Response**: We thank the reviewer for your valuable comments. We now include the definitions of the evaluation metrics in the Section 6 of the updated main manuscript (colored by blue). As a way to assess the interpretability of the learning model, Fig.2 gives a visual comparison of the energy predictions, but due to the limited space in the main manuscript, the numerical comparison of the energy predictions is shown in Table 2 in Appendix A.8.1. We apologize for the confusion caused by placing them in the appendix. We appreciate the reviewer's suggestion to include momentum error, but current work compares the predicted state derivatives to the true derivatives (RMSE in Table 1), which can reflect the momentum error. Furthermore, following the suggestion of Reviewer e4ns, we added an additional evaluation metric, the mean negative log likelihood (MNLL), which can evaluate the predicted uncertainty provided by the GP models. We invite the reviewer to browse the additional experimental results (colored by blue) with these metrics in Appendix A.8.3 of the updated draft.

---

> > > ### Author Response · Authors · 2023-11-22
> > > **Response to Reviewer uHv2 (Part-3 out of 3)**
> > >
> > > ---
> > > > **6.Comment (Q4)**: It is not clear how the input features will be employed for the present approach in a multi-degree of freedom system. Specifically, whether the approach is permutation invariant or not is not clear. That is, does the order in which the degrees of freedom are provided as the input matter or not? Authors should clarify.
> > >
> > > **Response**: We appreciate the reviewer’s question. To clarify, the multi-output GP model we proposed is not permutation invariant. If the model is trained with input features in a certain order, the order should be maintained when making predictions. This is because the matrix-valued kernel we used is constructed by computing the partial derivatives with respect to the input features, so the kernel associates each input feature with a specific position in the input vector, and changing the order of the features can lead to incorrect predictions. We understand that models with permutation invariance allows for greater flexibility and generalization in handling multi-degree of freedom systems. Although our current GP model is not permutation invariant, it is possible to enhance the model to have this property in the future work, refering to a recent work utilizing the intrinsic co-regionalization model to develop multi-output GPs with permutation invariant [B4].
> > >
> > > ---
> > >
> > > Thank you again for the constructive comments. We hope these explanations resolve the concerns. Any further questions or suggestions would be greatly appreciated.
> > >
> > > ### References
> > >
> > > [B1] Legaard, Christian, et al. *Constructing neural network based models for simulating dynamical systems*. ACM Computing Surveys 55.11 (2023): 1-34.
> > >
> > > [B2] Krishnapriyan, Aditi S., et al. *Learning continuous models for continuous physics*. Communications Physics 6.1 (2023): 319.
> > >
> > > [B3] Boya Hou, Sina Sanjari, et al. *Sparse Learning of Dynamical Systems in RKHS: An Operator-Theoretic Approach*. In International Conference on Machine Learning. PMLR, 2023
> > >
> > > [B4] Rønneberg, Leiv, et al. *Dose–response prediction for in-vitro drug combination datasets: a probabilistic approach*. BMC bioinformatics 24.1 (2023): 161.

---

### Official Review · Reviewer_Vrr3 · 2023-11-09

**Soundness:** 3 good
**Presentation:** 4 excellent
**Contribution:** 3 good
**Rating:** 6
**Confidence:** 3

**Summary:**

This manuscript proposes a Gaussian process regression model that incorporates the Helmholtz-Hodge decomposition and a method for eliminating indeterminacy by incorporating knowledge of symmetry into it as a constraint on the model. Compared to the baseline models, the proposed method has not only improved the predictive performance but also allows for the construction of interpretable models.

**Strengths:**

The strength of this manuscript lies in the fact that the interpretability of the nonparametric model, Gaussian process regression, was ensured by basing it on the Helmholtz Hodge decomposition, and that compensation was made for the identifiability of the estimated model in order to achieve a physical valid interpretation.

**Weaknesses:**

A weakness of this manuscript is the lack of discussion of interpretability in the demonstration experiments, despite the authors' claim that the proposed method has a high interpretability.
In addition, when the proposed method is applied to complex phenomena that often require interpretation, the symmetries of the system are often considered to be unknown, and in this case, it is considered to be difficult to achieve identifiability by the proposed method.
This point is also considered to pose difficulties in terms of improving the prediction performance by the proposed method.

**Questions:**

It would be better to add a demonstration for more complex systems with larger degrees of freedom.
It would be better to describe the concept of applying the proposed method to cases where the symmetry of the system is unknown.

---

> ### Author Response · Authors · 2023-11-22
> **Response to Reviewer Vrr3**
>
> We are very grateful for your valuable comments and acknowledging our main contributions. We greatly appreciate the time and effort you put into reviewing our paper. Below are our responses to your concerns and the changes we have made to address them.
>
> ---
>
> > **1. Comment**: A weakness of this manuscript is the lack of discussion of interpretability in the demonstration experiments, despite the authors' claim that the proposed method has a high interpretability.
>
> **Response**: We thank the reviewer for the opportunity to further clarify and demonstrate the interpretability of our model. We stated that our model is an interpretable way to learn a dynamical system because it allows one to learn the curl-free and div-free dynamics of a system separately, which are the two most ubiquitous differential invariants of vector fields in nature and are always physically meaningful. Therefore, our model can not only recover the dynamics of the full system, but also learn physically interpretable features that can provide new insights into the dynamical system. This interpretability is guaranteed by the identifiability of the model through symmetry constraints. To demonstrate this interpretability, the experiments detailed in Section 6.2 shows that the div-free features learned by the model with symmetry constraints (SPHHD-GP) can be used to accurately predict the energy evolution of dynamical systems. In contrast, models without symmetry constraints (HHD-GP and D-HNN) cannot learn the correct HHD decomposition due to model identifiability issues, so the features they learn are not physically interpretable. To further show the interpretability of our model, we add experiments of generalizing the learned models to predict dynamics with unseen friction coefficients, which demonstrate the interpretability of learned curl-free features. We invite the reviewer to view the experimental results in the Appendix A.8.4 (colored by blue) of the updated draft. We state that the interpretability of our model is guaranteed by the identifiability of the model achieved by symmetry constraints. Therefore, in addition to these experimental results, we have now added theoretical verification (Appendix A.6 in the updated draft) for the three demonstration examples that the non-identifiability of our model (i.e. non-uniqueness of the HHD) is solved through forced symmetries.
>
> ---
>
> > **2. Comment**: When the proposed method is applied to complex phenomena that often require interpretation, the symmetries of the system are often considered to be unknown, and in this case, it is considered to be difficult to achieve identifiability by the proposed method. This point is also considered to pose difficulties in terms of improving the prediction performance by the proposed method. It would be better to add a demonstration for more complex systems with larger degrees of freedom and describe the concept of applying the proposed method to cases where the symmetry of the system is unknown.
>
> **Response**: Comparative experiments between HHD-GP and SPHHD-GP (Symmetry-preserving HHD-GP) show that the interpretability of our model is difficult to guarantee when the symmetry is unknown. We understand the reviewer's concern about the availability of symmetry. Although physical systems always adhere to symmetries, the symmetries of many complex phenomena are often considered to be partially known or unknown. Considering the importance and ubiquitous of symmetry, there are a few recent works addressing this issue by learning hidden symmetries of unknown systems from data [A1, A2, A3]. This is an interesting direction but goes in a different direction from our main point, which is in finding ways to incorporate symmetries we already have into div-free and curl-free GPs and in demonstrating that symmetries are effective in improving the predictability and interpretability of GPs. Now that our model has been demonstrated to perform well when system symmetry is available, we look forward to enhancing our model with the capabilities of automatedly learning symmetries for the model in future work.
>
> ---
>
> Thank you again for the constructive comments. We hope these explanations resolve the concerns. Any further questions or suggestions would be greatly appreciated.
>
> ## References
>
> [A1] Liu, Ziming, et al. *Machine learning hidden symmetries*. Physical Review Letters 128.18 (2022): 180201.
>
> [A2] Desai, Krish, et al. *Symmetry discovery with deep learning*. Physical Review D 105.9 (2022): 096031.
>
> [A3] Forestano, Roy T., et al. *Deep learning symmetries and their Lie groups, algebras, and subalgebras from first principles*. Machine Learning: Science and Technology (2023).

---

### Meta-Review · Area_Chair_pf12 · 2023-12-06

**Metareview:**

This manuscript proposes a Gaussian process regression model that incorporates the Helmholtz-Hodge decomposition and addresses the underlying non-uniqueness by incorporating a symmetry constraint. Empirical studies on the damped mass-spring system, damped pendulum, and Chua circuit demonstrate improved performance compared to baseline models.

While reviewers acknowledge the novelty of the proposed method, there is a consensus that its applicability is limited due to high computational complexity, which also significantly restricts the dimensionality of the benchmark examples to toy problems, with baselines limited to only other Gaussian process models. Additionally, reviewers criticize the weak treatment of model interpretability, despite this being a major focus of the paper.

While the authors addressed some reviewer concerns in their responses, the core issues highlighted above remain unresolved.
Given the uncompelling, low-dimensional nature of the numerical experiments and the narrow selection of baselines, the advantages of this potentially interesting idea remain unclear and lackluster. Therefore, I recommend rejecting the paper.

**Justification For Why Not Higher Score:**

The experiments are low-dimensional toy models, which are not of great interest to the community at large. It is not clear how the method would scale to more realistic models, given the very high computational complexity.

**Justification For Why Not Lower Score:**

N/A

---

### Decision · Program_Chairs · 2024-01-16

Reject